# MEILB2-BRME1 forms a V-shaped DNA clamp upon BRCA2-binding in meiotic recombination

Manickam Gurusaran [1], Jingjing Zhang[2], Kexin Zhang[2], Hiroki Shibuya [2,3] & Owen R. Davies [1] ✉

DNA double-strand break repair by homologous recombination has a specialised role in meiosis by generating crossovers that enable the formation of haploid germ cells. This requires meiosis-specific MEILB2-BRME1, which interacts with BRCA2 to facilitate loading of recombinases onto resected DNA ends. Here, we report the crystal structure of the MEILB2-BRME1 2:2 core complex, revealing a parallel four-helical assembly that recruits BRME1 to meiotic double-strand breaks in vivo. It forms an N-terminal β-cap that binds to DNA, and a MEILB2 coiled-coil that bridges to C-terminal ARM domains. Upon BRCA2-binding, MEILB2-BRME1 2:2 complexes dimerize into a V-shaped 2:4:4 complex, with rod-like MEILB2-BRME1 components arranged at right-angles. The β-caps located at the tips of the MEILB2-BRME1 limbs are separated by 25 nm, allowing them to bridge between DNA molecules. Thus, we propose that BRCA2 induces MEILB2-BRME1 to function as a DNA clamp, connecting resected DNA ends or homologous chromosomes to facilitate meiotic recombination.

Breast cancer susceptibility protein BRCA2 performs a central role in recombination-mediated DNA double-strand breaks (DSBs) repair by loading recombinases onto resected DNA ends[1–3]. Its importance in genome stability is demonstrated by the gross chromosomal rearrangements that accumulate in BRCA2 deficient mammalian cells[4,5], and the strong association between germline BRCA2 mutations and early-onset breast and ovarian cancers[6]. BRCA2 is also required for meiosis, as its germline-specific deficiency leads to meiotic impairment and infertility in mice[7]. Here, programmed DSBs are induced, and then repaired by recombination between homologous chromosomes to generate crossovers, which provide genetic diversity and are critical for the correct segregation of homologues[8,9]. Hence, BRCA2 is essential for maintaining genome integrity in somatic cells and for generating haploid germ cells in meiosis.

In somatic cells, homologous recombination is initiated by DSB end resection, in which the MRN complex generates 3′ single-

stranded DNA (ssDNA) overhangs that become coated by RPA[10] (Fig. 1). BRCA2 then displaces RPA by loading recombinase RAD51 to form RAD51-ssDNA filaments[11,12]. These nucleoprotein filaments invade the sister chromatid and mediate template-directed repair, generating joint molecules that are resolved by multiple pathways[13]. In meiosis, homologous recombination uses the same principal machinery, with several critical modifications (Fig. 1): (1) meiotic DSBs are induced by the SPO11–TOPOVIBL complex[14,15]. (2) Resected ssDNA overhangs are coated by RPA and meiosis-specific complex MEIOB–SPATA22[16–18]. (3) BRCA2 loads RAD51 and the meiosis-specific recombinase DMC1 onto ssDNA overhangs[19–21]. (4) BRCA2 and recombinases are supported by essential meiosis-specific accessory proteins, including MEILB2–BRME1[22]. (5) Meiotic recombination uses the homologous chromosome rather than the sister chromatid as the primary repair template[23]. (6) Meiotic recombination occurs within the synaptonemal complex that binds together

[1]Wellcome Centre for Cell Biology, Institute of Cell Biology, University of Edinburgh, Edinburgh, UK. [2]Department of Chemistry and Molecular Biology, University of Gothenburg, Gothenburg, Sweden. [3]Laboratory for Gametogenesis, RIKEN Center for Biosystems Dynamics Research (BDR), Kobe, Hyogo, Japan. ✉e-mail: owen.davies@ed.ac.uk

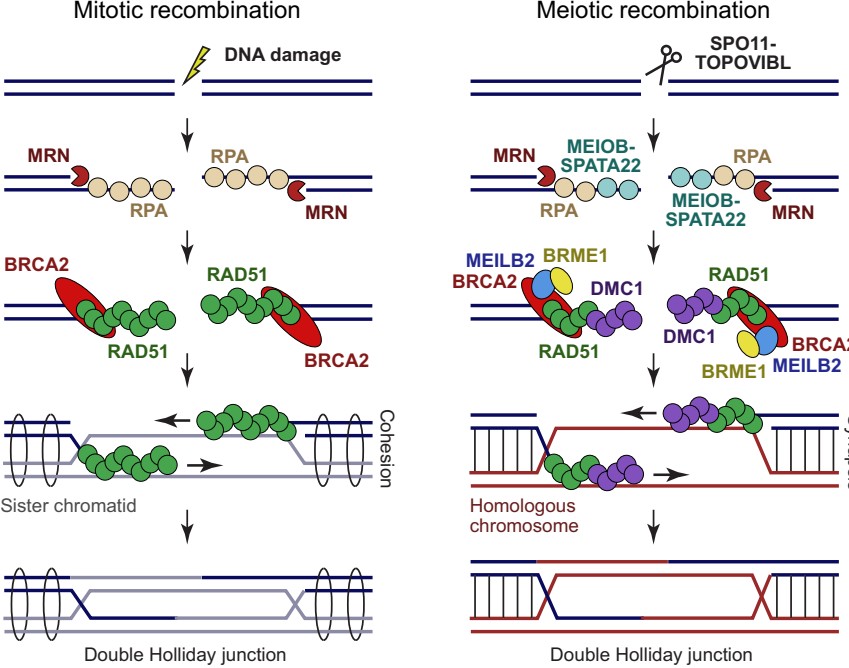

**Fig. 1 | Mechanics of homologous recombination.** Schematic of the key molecular players in mitotic (left) and meiotic (right) recombination. The main specialisations of meiotic recombination are DSB induction by SPO11–TOPOVIBL, coating of resected ends with MEIOB–SPATA22 (alongside RPA), the requirement for recombinase DMC1 (in addition to RAD51), and the role of MEILB2–BRME1 in BRCA2 function. These facilitate the use of the synapsed homologous chromosome, rather than the cohesin-bound sister chromatid, as the primary repair template. In meiotic recombination, double Holliday junctions are resolved into crossovers and non-crossovers in a highly regulated manner. In mitotic recombination, crossovers are less frequent, and additional pathways operate that bypass the double Holliday junction intermediate.

homologous chromosome axes[24]. (7) The processing of recombination intermediates is tightly regulated to ensure the formation of at least one crossover, and rarely more than two crossovers, per homologous chromosome pair[8,9].

BRCA2 is a large protein (3329 amino acids in mice) that contains multiple recombinase-binding sites and a single globular DNA-binding domain (Fig. 2a). BRC repeats within the central exon 11 region bind to RAD51 and DMC1 by competing with their oligomerisation[25,26]. In contrast, sequences that flank the DNA-binding domain (exons 14 and 27), interact with DMC1 and RAD51, respectively, in a manner that stabilises their nucleoprotein filaments[27–30]. Finally, the DNA-binding domain contains OB-folds for ssDNA-binding[31], and interacts with acidic DNA-mimicking protein DSS1[32], which act together to offload RPA from ssDNA overhangs[12,33]. Hence, whilst we lack a full molecular mechanism, BRCA2 contains the necessary functionalities to explain its role in displacing RPA, and loading recombinases onto ssDNA overhangs to form nucleoprotein filaments[1–3].

MEILB2 (meiotic localiser of BRCA2; also known as HSF2BP) was recently identified as a meiosis-specific interacting partner of BRCA2[34,35]. BRME1 (BRCA2 and MEILB2-associating protein 1; also known as MEIOK21, MMER, C19ORF57 and 4930432K21Rik) was then identified as a meiosis-specific binding partner of MEILB2[36–40]. MEILB2 and BRME2 co-localise on meiotic recombination sites in a mutually dependent manner[34,36]. In MEILB2 and BRME1 knockout mice, males are sterile, with failure of RAD51 and DMC1 foci formation, failure of DSB repair and failure of crossover formation[34–40]. In contrast, females exhibit subfertility, in which the development and number of oocytes is substantially reduced[34–41]. This resembles the sexual dimorphism observed upon deficiency of BRCA2 and other recombination factors[7]. Their co-localisation and similar phenotypes suggest that MEILB2 and BRME1 exist within the same functional unit in meiosis. This likely occurs through modulation of BRCA2 function as MEILB2 or BRME1 expression suppresses BRCA2-mediated recombination in somatic cells[36,42], and MEILB2 expression has been observed in cancer cell lines and human tumour samples[35,42]. Indeed, it has been hypothesised that MEILB2–BRME1 may modulate the oligomeric state or conformation of BRCA2 to favour meiotic rather than mitotic recombination[22].

The mechanical basis of recombination is an intrinsically structural problem that will likely be solved by understanding the functional architecture of its principal components and complexes. MEILB2 consists of N-terminal coiled-coil and C-terminal ARM (armadillo repeat) domains, which bind with high affinity to BRME1 and BRCA2, respectively[36,43] (Fig. 2a and Supplementary Fig. 1a). The ARM domains interact with the MEILB2-binding domain (MBD) of BRCA2, which is located immediately upstream of its DMC1-binding site[36] (Fig. 2a). Crystal structures have revealed that BRCA2-binding staples together two MEILB2-ARM dimers at ~95° to one another[43,44]. In these 2:4 complexes, each BRCA2 MBD peptide bridges between interacting MEILB2 ARM dimers, through cryptic repeats within their N- and C-terminal ends binding to opposing ARM domains[43,44]. The N-terminus of MEILB2's coiled-coil binds to the C-terminal end of BRME1[36] (Fig. 2a and Supplementary Fig. 1b), which currently has no other known structure or interacting partners. The MEILB2 coiled-coil exists in dimeric and higher order species in isolation, but forms stable 2:2 complexes with BRME1[36]. However, the lack of structural information regarding the MEILB2–BRME1 complex currently prevents us from understanding the architecture of the wider BRCA2–MEILB2–BRME1 assembly, and critically the molecular basis of MEILB2–BRME1 function in BRCA2-mediated meiotic recombination.

Here, we report the crystal structure of the MEILB2–BRME1 2:2 core complex, revealing a four-helical assembly that mediates BRME1 recruitment to meiotic DSBs in vivo. We show that its N-terminal β-cap binds to DNA, requiring both MEILB2 and BRME1 amino acids. We report a model of the full MEILB2–BRME1 2:2 complex, and show that BRCA2-binding induces dimerisation of this structure into a V-shaped assembly. The β-caps at the ends of the MEILB2–BRME1 limbs of this

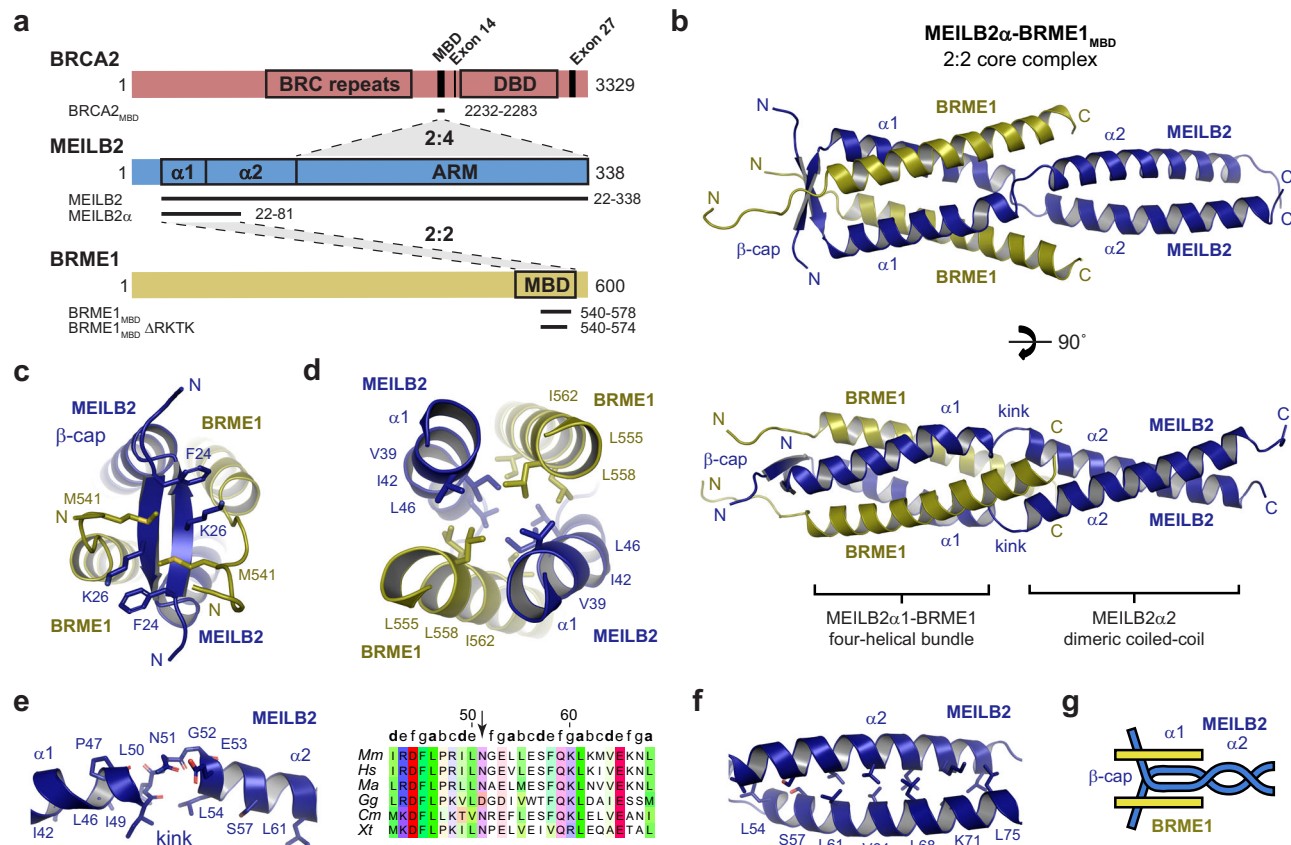

**Fig. 2 | Crystal structure of the MEILB2α–BRME1$_{MBD}$ 2:2 core complex.**
**a** Schematic of mouse BRCA2, MEILB2 and BRME1 sequences, highlighting domains, interaction sites and constructs used in this study. BRCA2 includes recombinase-binding BRC repeats, a MEILB2-binding domain (MBD), DMC1-binding exon 14, a DNA-binding domain (DBD) and RAD51-binding exon 27. MEILB2 contains a BRME1-binding α-helical region, and a BRCA2-binding armadillo repeat (ARM) domain. BRME1 has a MEILB2-binding domain (MBD). **b** Crystal structure of the MEILB2α–BRME1$_{MBD}$ 2:2 core complex. MEILB2 α1 (blue) and BRME1 (yellow) α-helices interact within a parallel four-helical bundle. MEILB2 α1 helices are followed by kinks that re-orientate α2 helices to form a parallel dimeric coiled-coil, and are preceded by a β-cap. **c** The N-terminal β-cap is a two-stranded anti-parallel β-sheet that is oriented perpendicular to the coiled-coil axis, supported by hydrophobic

interactions of MEILB2 F24 and K26 and BRME1 M541. **d** The hydrophobic core of the parallel four-helical bundle is formed of leucine, isoleucine and valine amino acids of MEILB2 and BRME1 chains, as indicated. **e** The kink within the MEILB2 chain is defined by non-helical backbone torsion angles of L50 and N51 (left), and a single amino-acid insertion between heptad repeats of α1 and α2 helices (indicated by an arrow, right). Multiple sequence alignment: *Mus musculus* (*Mm*), *Homo sapiens* (*Hs*), *Mesocricetus auratus* (*Ma*), *Gallus gallus* (*Gg*), *Callorhinchus milii* (*Cm*) and *Xenopus tropicalis* (*Xt*). The full alignment is shown in Supplementary Fig. 1a. **f** The MEILB2 α2 parallel dimeric coiled-coil is formed of 'a' and 'd' heptad amino-acids L54, S57, L61, V64, L68, K71 and L75. **g** Schematic of the MEILB2α–BRME1$_{MBD}$ 2:2 core structure.

structure are separated by 25 nm, so may bind together distinct DNA molecules, forming protein–DNA networks. Hence, we propose that BRCA2-binding may induce MEILB2–BRME1 to act as a DNA clamp in meiotic recombination.

## Results

### Crystal structure of the MEILB2α–BRME1$_{MBD}$ core complex

MEILB2 possesses an N-terminal α-helical region, consisting of two predicted α-helices (α1 and α2), followed by a C-terminal ARM domain (Fig. 2a). We previously demonstrated that MEILB2's α-helical region binds via its α1 helix to the C-terminus of BRME1 (MBD), forming a stable 2:2 complex[36]. Here, we identified an optimised construct for crystallography, which includes the α1 helix and the beginning of the α2 helix of MEILB2 (amino-acids 22–81; herein referred to as MEILB2α) and the core of BRME1's MBD (amino-acids 540–578; herein referred to as BRME1$_{MBD}$) (Supplementary Figs. 1 and 2). Crystals of this MEILB2α–BRME1$_{MBD}$ complex diffracted to a resolution limit of 1.50 Å, enabling structure solution by molecular replacement of ideal helical fragments using *ARCIMBOLDO_LITE*[45] (Table 1 and Supplementary Fig. 3). This revealed a 2:2 structure, consisting of an N-terminal

parallel four-helical bundle of MEILB2 α1 and BRME1, followed by a C-terminal MEILB2 α2 parallel dimeric coiled-coil (Fig. 2b).

The MEILB2α–BRME1$_{MBD}$ structure contains an unusual β-cap at its N-terminal tip (Fig. 2b, c). This consists of an anti-parallel two-stranded β-sheet, which is formed by the N-terminal ends of MEILB2 chains, and is orientated perpendicular to the helical axis (Fig. 2b, c). The β-cap binds together the N-termini of the two α1 helices of the four-helical bundle, so likely prevents the fraying apart of helical ends that typically occurs within coiled-coil structures[46]. Hence, we infer that the β-cap likely confers rigidity to the N-terminal end of the MEILB2α–BRME1$_{MBD}$ 2:2 structure. Further, the β-cap shows a high level of evolutionary conservation that is comparable to MEILB2's α1 and ARM regions, suggesting that it is important for function (Supplementary Fig. 1a). In previous cases of α/β-coiled-coils, the β-sheets typically constitute insertions, as β-layers within the α-helical coiled-coil structure[47]. For example, PDB accession 2BA2 includes a β-layer, which is similar to the β-cap, but separates rather than capping coiled-coils[48]. To our knowledge, MEILB2α–BRME1$_{MBD}$ is the first example of an α/β-coiled-coil in which the β-sheet caps off the end of the α-helical coiled-coil.

**Table 1 | Data collection, phasing and refinement statistics**

| | MEILB2α–BRME1$_{MBD}$ 2:2 core complex |
|---|---|
| PDB accession | 7Z8Z |
| Data collection | |
| Space group | P2$_1$2$_1$2$_1$ |
| Cell dimensions | |
| a, b, c (Å) | 27.05, 43.99, 178.71 |
| α, β, γ (°) | 90, 90, 90 |
| Resolution (Å) | 89.36–1.50 (1.53–1.50)$^a$ |
| $R_{meas}$ | 0.130 (1.016) |
| $R_{pim}$ | 0.020 (0.309) |
| $I/\sigma(I)$ | 16.7 (2.2) |
| $CC_{1/2}$ | 0.999 (0.942) |
| Completeness (%) | 99.1 (94.7) |
| Redundancy | 35.2 (10.2) |
| Refinement | |
| Resolution (Å) | 44.68–1.50 |
| No. of reflections | 34,833 |
| $R_{work}/R_{free}$ | 0.1965/0.2347 |
| No. of atoms | 1896 |
| Protein | 1584 |
| Ligand/ion | 0 |
| Water | 312 |
| B-factors | 36.94 |
| Protein | 37.07 |
| Ligand/ion | N/A |
| Water | 36.24 |
| R.m.s. deviations | |
| Bond lengths (Å) | 0.0045 |
| Bond angles (°) | 0.623 |

$^a$Values in parentheses are for highest-resolution shell.

Following the β-cap, a MEILB2–BRME1 four-helical bundle is observed, terminating with the splaying apart of the C-termini of BRME1 chains (Fig. 2b, d). At this point, there is a slight 'kink' in the MEILB2 chains, between α1 and α2 helices, where L50 and N51 adopt non-helical conformations due to a single amino-acid insertion into the heptad repeats (Fig. 2e). This kink re-orientates the MEILB2 chains from the upstream four-helical bundle to the downstream dimeric coiled-coil conformation (Fig. 2b, e). The parallel dimeric coiled-coil of the α2 helix adopts a canonical heptad pattern (Fig. 2b, f). Thus, the overall structure of MEILB2α–BRME1$_{MBD}$ can be described as a parallel 2:2 α/β-coiled-coil, where an N-terminal MEILB2 β-cap is followed by a MEILB2α1–BRME1 four-helical bundle that transitions via a kink into a C-terminal MEILB2 α2 dimeric coiled-coil (Fig. 2b, g).

**BRME1 recruitment to meiotic DSBs requires its MEILB2-binding interface**

To determine whether the MEILB2–BRME1 interaction observed in the crystal structure is crucial for their interaction in vivo, we generated a point mutant of BRME1 that specifically targets its MEILB2-binding site. The hydrophobic core of the four-helical bundle comprises highly conserved BRME1 amino acids V548, L555 and I562, which occupy the 'a' positions within the heptad repeats (Fig. 3a and Supplementary Fig. 1b). Therefore, we introduced glutamate mutations at these residues (V548E, L555E and I562E; hereafter referred to as 3E) to disrupt the assembly of the four-helical bundle. We confirmed this disruption biochemically, demonstrating that the 3E mutation effectively

prevented the formation of the MEILB2α–BRME1$_{MBD}$ complex in vitro (Fig. 3b and Supplementary Fig. 4).

In a previous study, we established that GFP–BRME1 is recruited to meiotic DSBs upon expression in mouse spermatocytes through in vivo electroporation[36]. Therefore, we employed the same system to investigate the impact of the 3E mutation on the localisation of full-length BRME1. While GFP–BRME1$_{FL}$ 3E was expressed at a comparable level to the wild-type protein (Fig. 3c), it failed to be recruited to meiotic DSBs in zygotene and pachytene spermatocytes (Fig. 3d). These findings confirm that the MEILB2-binding interface of BRME1 observed in the crystal structure is solely responsible for its interaction with MEILB2 and its recruitment to meiotic DSBs in vivo.

**Structure of the full MEILB2–BRME1$_{MBD}$ complex**

Our MEILB2α–BRME1$_{MBD}$ crystal structure contains the N-terminal end of the MEILB2 α2 coiled-coil, up to amino-acid Q77 (Fig. 2b). The C-terminal end of the same coiled-coil, from the equivalent of amino-acid Q109, was observed in previous crystal structures of the BRCA2$_{MBD}$–MEILB2 ARM complex (PDB accessions 7LDG and 7BDX)[43,44]. Hence, MEILB2 α2 likely forms a continuous coiled-coil that physically separates the BRME1- and BRCA2-binding regions of MEILB2. To visualise this, and build the intervening 31 residues, we generated *AlphaFold2* models of the 2:2 complex between the full structured region of MEILB2 (amino-acids 22–338; herein referred to as MEILB2) and BRME1[49,50]. We chose to remove the N-terminal 21 amino acids from MEILB2 as they are poorly conserved and are predicted to be unstructured, so would interfere with subsequent biophysical studies (Supplementary Fig. 1a). During modelling, we enforced the use of our MEILB2α–BRME1$_{MBD}$ crystal structure and the previous BRCA2$_{MBD}$–MEILB2 ARM structure (PDB accession 7LDG)[43] as the sole templates. The resultant model demonstrates that the MEILB2 α2 coiled-coil can seamlessly connect the BRME1- and BRCA2-binding regions of the MEILB2 dimer (Fig. 4a and Supplementary Fig. 5). Further, it predicts an overall length of ~18 nm between the N-terminal β-cap and the C-terminal ARM domains.

We confirmed the validity of the MEILB2–BRME1$_{MBD}$ 2:2 model by size-exclusion chromatography small-angle X-ray scattering (SEC-SAXS) analysis of the MEILB2–BRME1$_{MBD}$ complex in solution. The SAXS real-space (r) pair-distance distribution indicated a rod-like molecule of 20-nm length, including a 14-nm separation between domains, consistent with the length and location of four-helical bundle and ARM domains within the model (Fig. 4b, Supplementary Fig. 6a–c and Supplementary Table 1). Further, the SAXS scattering curve was closely fitted by the MEILB2–BRME1$_{MBD}$ model, similar to the fit between the SAXS data and crystal structure of the core complex, at $\chi^2$ values of 2.96 and 2.50, respectively (Fig. 4b). To test the specificity of SAXS data fitting, we generated a deformed version of the MEILB2–BRME1$_{MBD}$ model in which MEILB2 α1 helices and BRME1$_{MBD}$ chains were folded back onto the α2 coiled-coil to create a structurally similar but shortened model (Supplementary Fig. 6d). The deformed model gave large residuals, with a $\chi^2$ value of 21.08 (Fig. 4b). Hence, fitting to the MEILB2–BRME1$_{MBD}$ SAXS data is sensitive to small structural changes, providing confidence that the close fit of the MEILB2–BRME1$_{MBD}$ model is specific and indicative that this model accurately reflects the solution structure. Thus, we conclude that MEILB2–BRME1$_{MBD}$ has a linear structure in which the N-terminal BRME1-binding core is connected to C-terminal ARM domains by a MEILB2 α2 coiled-coil (Fig. 4c).

**MEILB2–BRME1$_{MBD}$ binds to DNA via its β-cap**

The surface electrostatics of the MEILB2–BRME1$_{MBD}$ 2:2 model indicated that the N-terminal MEILB2–BRME1$_{MBD}$ core contains discrete patches of basic charge, whereas the C-terminal ARM domains are predominantly acidic (Fig. 5a). Hence, we wondered whether the N-terminal end of the molecule may bind to DNA. We noticed that addition of dsDNA led to the

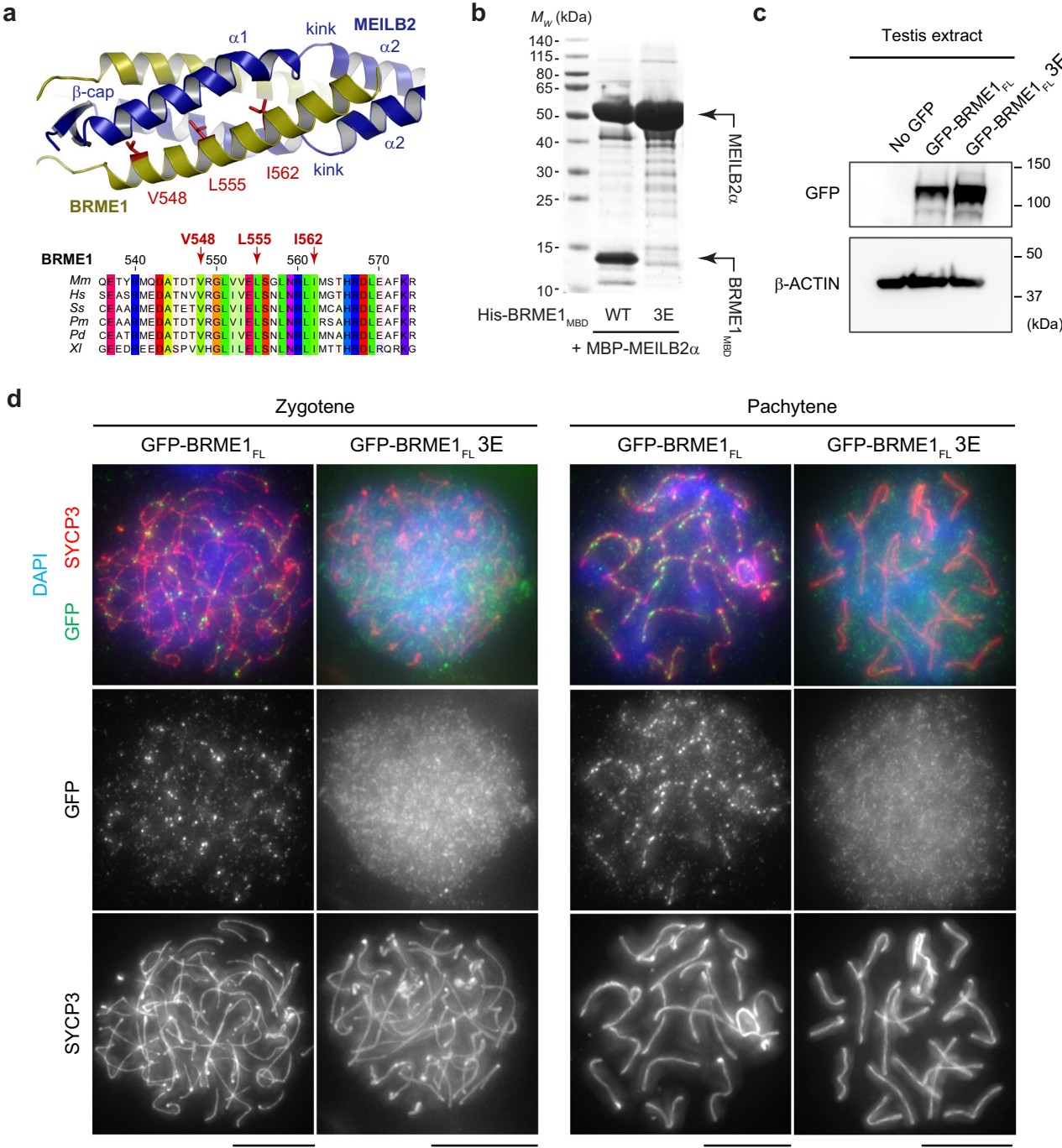

**Fig. 3 | The MEILB2-binding interface is required for BRME1 recruitment to meiotic DSBs. a** The MEILB2–BRME1 interaction involves BRME1 amino-acids V548, L555 and I562, which are buried in the hydrophobic core of the four-helical bundle (top), and are evolutionarily conserved (bottom). Multiple sequence alignment: *Mus musculus* (*Mm*), *Homo sapiens* (*Hs*), *Sus scrofa* (*Ss*), *Physeter macrocephalus* (*Pm*), *Phyllostomus discolour* (*Pd*) and *Xenopus laevis* (*Xl*). The full alignment is shown in Supplementary Fig. 1b. **b** Amylose pulldown of His-BRME1$_{MBD}$ with MBP–MEILB2α following recombinant co-expression. MEILB2-binding of BRME1 wildtype (WT) is blocked by the BRME1 V548E L555E I562E (3E) mutation; representative of three independent experiments. This SDS–PAGE shows the amylose precipitation of MEILB2–BRME1$_{MBD}$ at a physiological salt concentration

elution fractions; all other fractions of the same experiment are shown in Supplementary Fig. 4. Expression of GFP–BRME1$_{FL}$ (full-length, wildtype) and GFP–BRME1$_{FL}$ 3E (full-length, 3E mutant) in mouse spermatocytes by in vivo electroporation; representative of three independent experiments. **c** Immunoblots of testis extracts following electroporation with GFP–BRME1$_{FL}$ and GFP–BRME1$_{FL}$ 3E, blotted with antibodies against GFP (top) and β-Actin (bottom). **d** Mouse zygotene (left) and pachytene (right) spermatocytes expressing GFP–BRME1$_{FL}$ and GFP–BRME1$_{FL}$ 3E, stained with anti-SYCP3 antibody (red), anti-GFP antibody (green), and 4,6-diamidino-2-phenylindole (DAPI). Scale bars, 5 μm. Source data are provided as a Source data file.

of 150 mM KCl, whereas the isolated protein remained soluble in these conditions (Fig. 5b). Further, using streptavidin magnetic beads, we demonstrated that a biotinylated dsDNA substrate could pulldown the MEILB2–BRME1$_{MBD}$ complex at 150 mM KCl (Fig. 5c). These observations suggest that MEILB2–BRME1$_{MBD}$ is a DNA-binding protein.

The DNA-induced precipitation of MEILB2–BRME1$_{MBD}$ at physiological salt concentrations precluded its study by solution biophysics

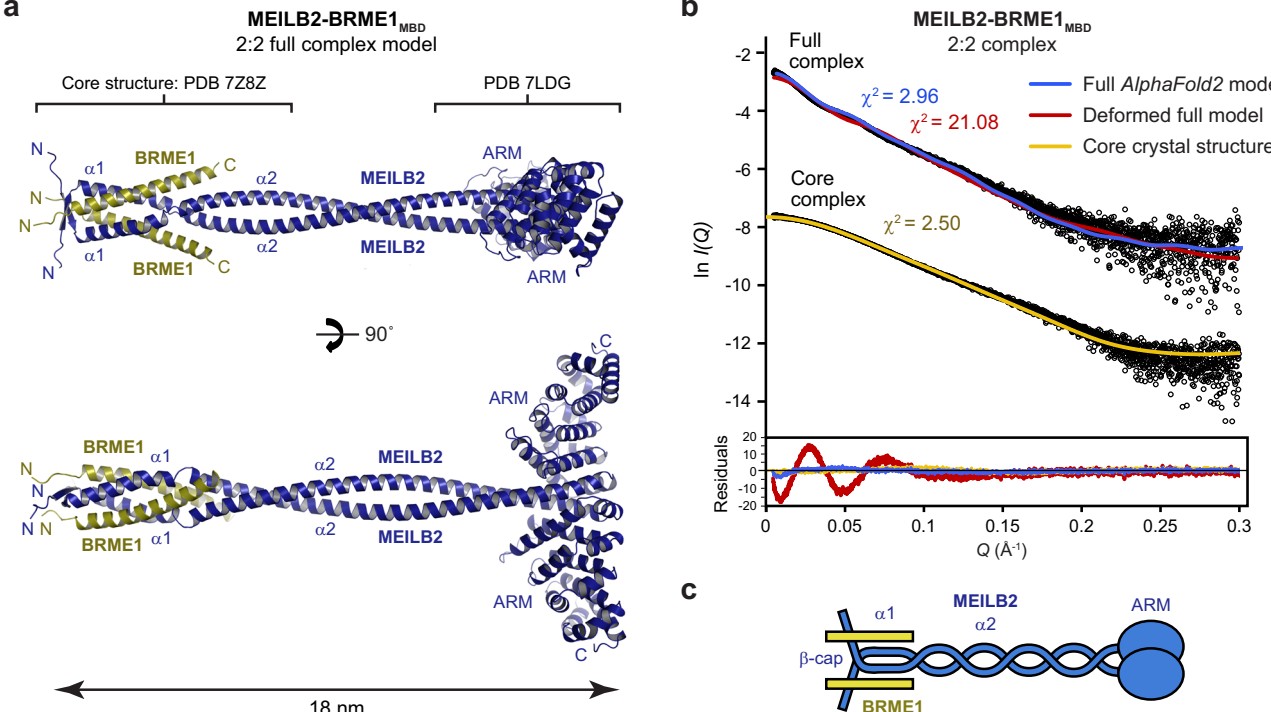

**Fig. 4 | Structure of the full MEILB2–BRME1_MBD 2:2 complex. a** *AlphaFold2* model of the full MEILB2–BRME1_MBD 2:2 complex, generated using the MEILB2α–BRME1_MBD 2:2 structure reported herein (PDB accession 7Z8Z), and the MEILB2 ARM domain structure (PDB accession 7LDG)[43], as the sole templates. The molecule has a length of ~18 nm, including a 14-nm separation between the N-terminal four-helical bundle and the C-terminal ARM domains. Modelling details, scores and plots are provided in Supplementary Fig. 5. **b** SEC-SAXS scattering curves of the MEILB2–BRME1_MBD full complex and the MEILB2α–BRME1_MBD core complex, overlaid with the theoretical scattering curves of the *AlphaFold2* model (blue), a deformed and shortened version of the *AlphaFold2* model (red; Supplementary Fig. 6d), and the core crystal structure (yellow), with $\chi^2$ values of 2.96, 21.08 and 2.50, respectively. Residuals for each fit are shown (inset). Guinier analyses and real-space $P(r)$ distributions are shown in Supplementary Fig. 6. **c** Schematic of the MEILB2–BRME1_MBD 2:2 structure.

methods. Instead, we used electrophoretic mobility shift assays (EMSAs) to visualise discrete protein–DNA species by loading samples at 325 mM KCl and relying on gradual salt reduction as the complex enters the gel to prevent precipitation. Using this method, we demonstrated that MEILB2–BRME1_MBD can form discrete protein–DNA complexes with dsDNA, poly-dT ssDNA and ssDNA–dsDNA junctional substrates (Fig. 5d, e and Supplementary Fig. 7a). In addition to discrete species, some material remained the wells, likely due to the same process that led to precipitation in solution (Fig. 5d, e and Supplementary Fig. 7a). DNA binding showed no overt sequence specificity or preference for dsDNA, ssDNA or ssDNA–dsDNA junctional substrates. Further, saturation occurred at ratios of one MEILB2–BRME1_MBD 2:2 complex to 10 base pairs (dsDNA) or 20 bases (ssDNA) (Fig. 5d, e). Quantification of EMSAs conducted with substrate concentrations of 50 and 100 nM, and fitting to the Hill equation, revealed positive cooperativity with binding affinities of $520 \pm 60$ nM and $870 \pm 70$ nM for dsDNA and ssDNA, respectively (Fig. 5f). These findings are consistent with MEILB2–BRME1_MBD binding to the DNA backbone, with a binding footprint spanning ten base pairs. On this basis, we chose to use dsDNA substrates in subsequent DNA-binding analyses.

Notably, the basic charge of MEILB2–BRME1_MBD is particularly concentrated at its N-terminal β-cap (Fig. 5a). Given that the β-cap stabilises the end of the coiled-coil, we hypothesised that it may serve as a rigid platform for DNA binding at the tip of the MEILB2–BRME1 complex. To test this, we introduced glutamate mutations of MEILB2 amino-acid K26, and BRME1 amino-acids R540 and R549, which are the main contributors of basic charge at the β-cap (Fig. 5g). In both cases, the mutations did not interfere with the formation of the MEILB2–BRME1_MBD 2:2 complex (Supplementary Figs. 2 and 7b, c). The MEILB2 K26E mutation largely abrogated DNA-binding of

MEILB2–BRME1_MBD by EMSA, retaining only a slight hint of binding at the highest concentration (Fig. 5h). The BRME1 R540E R549E mutation completely abolished DNA binding of MEILB2–BRME1_MBD by EMSA (Fig. 5i). Further, this mutation also abrogated DNA-induced precipitation at 150 mM KCl, and abolished pulldown by biotinylated DNA (Fig. 5b, c). Thus, DNA binding is mediated by the rigid β-cap at the N-terminal tip of the MEILB2–BRME1_MBD structure (Fig. 5j). Further, MEILB2 failed to bind DNA in the absence of BRME1_MBD (Supplementary Fig. 7d). Hence, DNA binding involves amino acids from both MEILB2 and BRME1, and appears to be a consequence of complex formation rather than being an intrinsic property of either component.

## BRCA2_MBD dimerises MEILB2–BRME1_MBD to form a V-shaped assembly

What is the structure of the complex formed by MEILB2–BRME1_MBD upon interaction with BRCA2_MBD? It was previously shown that BRCA2_MBD-binding dimerises ARM domains from opposing MEILB2 dimers to form a stable 2:4 complex[43,44]. To test whether this also occurs in the presence of BRME1_MBD, we used size-exclusion chromatography multi-angle light scattering (SEC-MALS) to determine the oligomeric state of MEILB2–BRME1_MBD upon binding to BRCA2's MBD (amino-acids 2232–2283; herein referred to as BRCA2_MBD). This revealed that MEILB2–BRME1_MBD and BRCA2_MBD–MEILB2–BRME1_MBD are stable 2:2 and 2:4:4 complexes, respectively (Fig. 6a and Supplementary Figs. 2 and 8a, b). We also analysed BRCA2_MBD–MEILB2–BRME1_MBD in which MEILB2 and BRCA2_MBD harboured N-terminal MBP-tags, confirming its 2:4:4 oligomeric state (Supplementary Fig. 8c). Hence, BRCA2-binding induces dimerisation of MEILB2–BRME1 to form a stable 2:4:4 ternary complex.

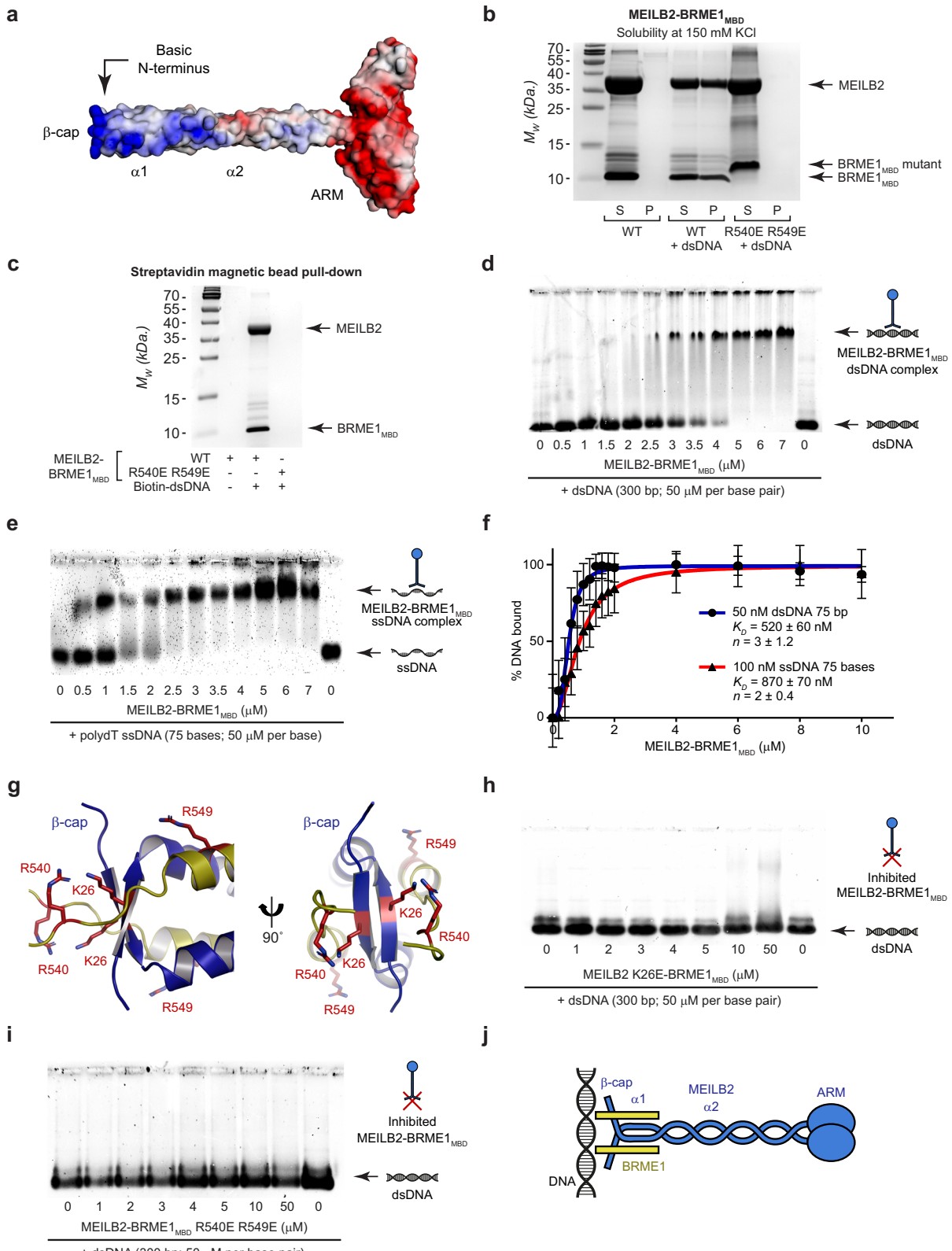

We next modelled the BRCA2$_{MBD}$−MEILB2−BRME1$_{MBD}$ 2:4:4 structure by docking two MEILB2−BRME1$_{MBD}$ models onto the constituent MEILB2 ARM dimers of the previous BRCA2$_{MBD}$−MEILB2 ARM 2:4 crystal structure (PDB accession 7LDG)[43] (Supplementary Fig. 9). The resultant model reveals a V-shaped assembly in which MEILB2−BRME1$_{MBD}$ 2:2 complexes are held at an angle of ~95° to one another, with their opposing ARM domains stapled together by BRCA2

MBDs (Fig. 6b). The unusual architecture of this assembly imposes a separation of ~25 nm between the β-caps at the tips of its limbs (Fig. 6b). We validated this model by cryo-EM and SEC-SAXS.

First, cryo-EM reference-free 2D class averages revealed V-shaped structures in which two limbs are bound together by a globular ring-like core (Fig. 6c). These 2D class averages are consistent with the expected appearance of the same V-shaped structure when viewed in

**Fig. 5 | DNA binding is mediated by the β-cap of MEILB2–BRME1$_{MBD}$. a** Surface electrostatic potential of the MEILB2–BRME1$_{MBD}$ modelled structure (red, electronegative; blue, electropositive). **b** SDS–PAGE of the supernatant (s) and pellet (p) following incubation of 30 μM MEILB2–BRME1$_{MBD}$ wildtype (WT) and BRME1 R540E R549E mutation at 150 mM KCl in the presence or absence of 1 μM (per molecule) dsDNA (75 base pairs); representative of three independent experiments. **c** Pulldown using streptavidin magnetic beads of MEILB2–BRME1$_{MBD}$ wildtype (WT) and BRME1 R540E R549E mutation in the presence or absence of a biotinylated dsDNA (75 base pairs) substrate; representative of three independent experiments. Electrophoretic mobility shift assays (EMSAs) analysing the ability of MEILB2–BRME1 (at molecular concentrations indicated) to interact with **d** dsDNA (300 base pairs) and **e** poly-dT ssDNA (75 bases), at concentrations of 50 μM per base/base pair. **f** Quantification of DNA binding by MEILB2–BRME1 through densitometry of EMSAs performed using 50 nM (per molecule) FAM-dsDNA (75 base pairs; blue) and 100 nM (per molecule) FAM-ssDNA (75 nucleotides; red). Plots, $K_D$ and Hill coefficient (*n*) values were determined by fitting data to the Hill equation and are quoted within a 95% confidence interval; data are presented as mean values, with error bars indicating standard error, *n* = 3 EMSAs. **g** The N-terminal β-cap of the MEILB2α–BRME1$_{MBD}$ 2:2 core structure includes basic amino-acids MEILB2 K26 and BRME1 R540 and R549. EMSAs analysing the ability of MEILB2–BRME1$_{MBD}$ (at molecular concentrations indicated) harbouring **h** MEILB2 K26E and **i** BRME1 R540E R549E mutations to interact with dsDNA (300 base pair) at concentrations of 50 μM per base pair. **d**, **e**, **h**, **i** Gel images are representative of at least three replicates performed using different protein samples. **j** Schematic of MEILB2–BRME1$_{MBD}$ interacting via its β-cap with DNA. Source data are provided as a Source Data file.

different orientations. Further, we generated an ab initio 3D model from the cryo-EM data, demonstrating a V-shaped structure, with an approximate angle of 100° between limbs and 25 nm separation between the β-caps at the tips of its limbs (Fig. 6d). The appearance and dimensions of the cryo-EM ab initio 3D model match those of our BRCA2$_{MBD}$–MEILB2–BRME1$_{MBD}$ 2:4:4 model. Second, SAXS analysis confirmed that BRCA2$_{MBD}$–MEILB2–BRME1$_{MBD}$ is an elongated molecule of up to 27 nm in length, consistent with the distance between β-caps, and the SAXS scattering curve was closely fitted by the 2:4:4 model ($χ^2$ = 1.36; Fig. 6e, Supplementary Fig. 10 and Supplementary Table 1). Hence, BRCA2$_{MBD}$-binding induces dimerisation of MEILB2–BRME1$_{MBD}$ 2:2 complexes into a V-shaped 2:4:4 assembly, where its two limbs are held at slightly greater than right angles to one another, and the β-caps at their N-terminal tips are separated by ~25 nm (Fig. 6f).

## BRCA2$_{MBD}$–MEILB2–BRME1$_{MBD}$ acts as a DNA clamp

The physical separation between the two DNA-binding β-caps of the BRCA2$_{MBD}$–MEILB2–BRME1$_{MBD}$ 2:4:4 ternary complex suggested that it may be able to bridge between DNA molecules. BRCA2$_{MBD}$–MEILB2–BRME1$_{MBD}$ was stable at 325 mM KCl, but precipitated at a physiological salt concentration of 150 mM KCl, even in the absence of DNA. This precluded analysis of DNA binding by solution biophysics methods. Hence, we sought to visualise protein–DNA species by EMSAs, which we had previously used to overcome the DNA-induced precipitation of MEILB2–BRME1$_{MBD}$. This revealed that BRCA2$_{MBD}$–MEILB2–BRME1$_{MBD}$ readily bound to a 75 base pair dsDNA substrate, forming large-scale protein–DNA structures that mostly remained in the well (Fig. 7a). We observed no overt difference in binding nature and affinity between dsDNA, ssDNA and ssDNA/dsDNA junctional substrates (Fig. 7a and Supplementary Fig. 11a–c), suggesting that binding likely occurs through the DNA backbone. Further, barely any DNA binding was observed for a BRCA2$_{MBD}$–MEILB2 complex in the absence of BRME1$_{MBD}$, indicating that it is dependent on the formation of the ternary complex (Supplementary Fig. 11d). As it is difficult to interpret the nature of large-scale protein–DNA structures that remain in the well, we additionally analysed a truncated BRCA2$_{MBD}$–MEILB2–BRME1$_{MBD}$ complex that tempered DNA binding and reduced precipitation. The truncation removed basic C-terminal amino acids of BRME1 (deletion of 575-RKTK-579; herein referred to as BRME1$_{MBD}$ ΔRKTK), and did not interfere with MEILB2-binding (Supplementary Fig. 12a, b). The BRCA2$_{MBD}$–MEILB2–BRME1$_{MBD}$ ΔRKTK complex formed a large protein–DNA species that was successfully resolved by EMSA, and substantially reduced the amount of material retained in the well (Fig. 7b). Hence, we used the truncated ΔRKTK complex to assist in determining the nature of DNA binding by the BRCA2$_{MBD}$–MEILB2–BRME1$_{MBD}$ ternary complex.

We reasoned that if the DNA substrate were of sufficient length, it should be possible for the two β-caps of the ternary complex to bind cooperatively to the same DNA molecule. Hence, we analysed a 300 base pair substrate, which has a length of 90 nm that far exceeds the 25 nm gap between β-caps. EMSAs showed that BRCA2$_{MBD}$–MEILB2–BRME1$_{MBD}$ initially formed a large resolved species with the 300 base pair substrate, and then generated larger structures that remained in the well when present at stoichiometric excess (Fig. 7c). This effect was particularly pronounced for the BRCA2$_{MBD}$–MEILB2–BRME1$_{MBD}$ ΔRKTK complex, which formed small discrete species prior to assembly into larger resolved and then unresolved structures at higher ratios (Fig. 7d). These small discrete species were not present for the 75 base pair substrate (Fig. 7a, b), which has an approximate length of 25.5 nm, so is insufficient to bind and span between the two DNA-binding β-caps. Hence, the small discrete species may represent BRCA2$_{MBD}$–MEILB2–BRME1$_{MBD}$ complexes bound cooperatively via both β-caps to the same DNA molecule. In contrast, the large-scale structures observed for both DNA substrates may include protein–DNA networks in which the two DNA-binding β-caps of BRCA2$_{MBD}$–MEILB2–BRME1$_{MBD}$ bridge between DNA molecules. Whilst the large resolved species may be ordered protein–DNA networks, the larger species retained in the well may be more complicated assemblies that could involve both bridging interactions and the process responsible for precipitation in solution.

The BRME1 R540 R549E mutation abrogated DNA binding of the BRCA2$_{MBD}$–MEILB2–BRME1$_{MBD}$ complex (Fig. 7e). As the mutated complex still precipitated at 150 mM KCl, this confirms that the DNA binding observed for the ternary complex is specific and is mediated by the β-caps. The MEILB2 K26E mutation also blocked binding of BRCA2$_{MBD}$–MEILB2–BRME1$_{MBD}$ to the shorter 75 base pair substrate (Fig. 7f). However, it retained DNA binding to the longer 300 base pair substrate, albeit forming a single discrete species rather than large-scale protein–DNA structures (Fig. 7g). Importantly, the stability and 2:4:4 oligomeric state of the BRCA2$_{MBD}$–MEILB2–BRME1$_{MBD}$ complex were not affected by the MEILB2 K26E mutation (Supplementary Figs. 2 and 12c). Hence, whilst the MEILB2 K26E mutation was sufficient to block DNA binding by a single β-cap, it likely retained sufficient residual affinity for the two β-caps of the V-shaped ternary complex to interact cooperatively with a sufficiently long single DNA molecule. Thus, the MEILB2 K26E mutant provides additional evidence in favour of our model.

Finally, we visualised complexes between BRCA2$_{MBD}$–MEILB2–BRME1$_{MBD}$ and dsDNA by atomic force microscopy (AFM). Whilst free dsDNA molecules were clearly separated, BRCA2$_{MBD}$–MEILB2–BRME1$_{MBD}$ formed large assemblies on dsDNA molecules and induced their tangling and looping (Fig. 7h). These observations are consistent with our model in which V-shaped BRCA2$_{MBD}$–MEILB2–BRME1$_{MBD}$ complexes tether together DNA molecules through their β-cap DNA-binding sites. Importantly, we did not observe tangled or looped dsDNA when using BRCA2$_{MBD}$–MEILB2–BRME1$_{MBD}$ complexes harbouring BRME1 R540E R549E mutation (Supplementary Fig. 13). Hence, our combined EMSA, AFM and mutant data support a model in which the two β-caps of the BRCA2$_{MBD}$–MEILB2–BRME1$_{MBD}$ complex act as spatially separated DNA-binding sites that can bridge between discrete DNA molecules.

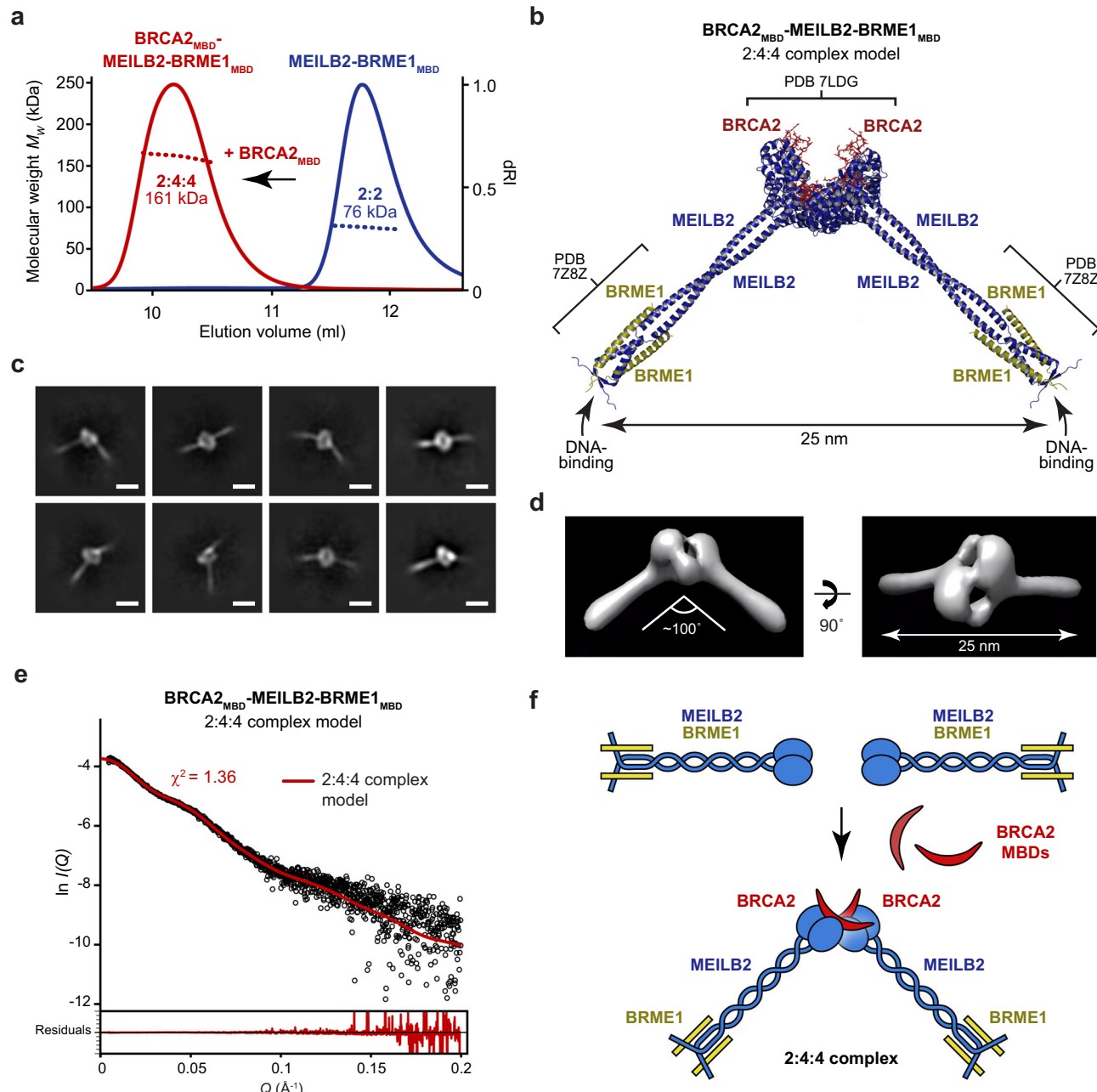

**Fig. 6 | Structure of the BRCA2$_{MBD}$–MEILB2–BRME1$_{MBD}$ 2:4:4 ternary complex.**
**a** SEC-MALS analysis of MEILB2–BRME1$_{MBD}$ (blue) and the BRCA2$_{MBD}$–MEILB2–BRME1$_{MBD}$ ternary complex (red), showing differential refractive index (dRI; solid lines) profiles with fitted molecular weights ($M_w$; dashed lines) across elution peaks. MEILB2–BRME1$_{MBD}$ forms a 76 kDa 2:2 complex (theoretical $M_w$–82 kDa), and BRCA2$_{MBD}$–MEILB2–BRME1$_{MBD}$ forms a 161 kDa 2:4:4 complex (theoretical $M_w$–176 kDa). SDS–PAGE of elution fractions are shown in Supplementary Fig. 8a, b. **b** Model of the BRCA2$_{MBD}$–MEILB2–BRME1$_{MBD}$ 2:4:4 complex generated by docking two MEILB2–BRME1$_{MBD}$ 2:2 complex models (shown in Fig. 4a) onto the BRCA2$_{MBD}$–MEILB2 ARM domain 2:4 complex crystal structure PDB accession 7LDG[43]. The model predicts that the DNA-binding β-caps of the two constituent MEILB2α–BRME1$_{MBD}$ core complexes are separated by 25 nm. Modelling details are shown in Supplementary Fig. 9. **c** Cryo-EM reference-free 2D class averages of the BRCA2$_{MBD}$–MEILB2–BRME1$_{MBD}$ 2:4:4 complex (21,175 particles). Scale bars, 10 nm. **d** Cryo-EM ab initio 3D model of the BRCA2$_{MBD}$–MEILB2–BRME1$_{MBD}$ 2:4:4 complex (13,027 particles; GSFSC = 13 Å), indicating an angle of ~100° between limbs and a 25 nm distance between β-caps at the N-terminal ends of its limbs. **e** SEC-SAXS data in which the BRCA2$_{MBD}$–MEILB2–BRME1$_{MBD}$ scattering curve is overlaid with the theoretical scattering curve of the 2:4:4 modelled structure (red; shown in (**b**)), with a $\chi^2$ value of 1.36. Residuals are shown (inset). Guinier analysis and real-space $P(r)$ distribution are shown in Supplementary Fig. 10. **f** Schematic of BRCA2$_{MBD}$–MEILB2–BRME1$_{MBD}$ complex formation, in which BRCA2-binding induces dimerisation of two MEILB2–BRME1$_{MBD}$ 2:2 complexes into a V-shaped 2:4:4 complex.

In summary, BRCA2$_{MBD}$-binding induces dimerisation of MEILB2–BRME1$_{MBD}$ into a V-shaped 2:4:4 assembly, in which DNA-binding β-caps at the tips of its two limbs can independently interact with DNA molecules. Hence, BRCA2$_{MBD}$ induces MEILB2–BRME1$_{MBD}$ to act as a DNA clamp that holds together discrete DNA molecules, with a separation of ~25 nm (Fig. 8). Thus, we propose that within the context of their full-length molecules, BRCA2 may induce the clamping together of DNA molecules by MEILB2–BRME1 to facilitate inter-homologue recombination as part of the specialised mechanics of meiotic recombination.

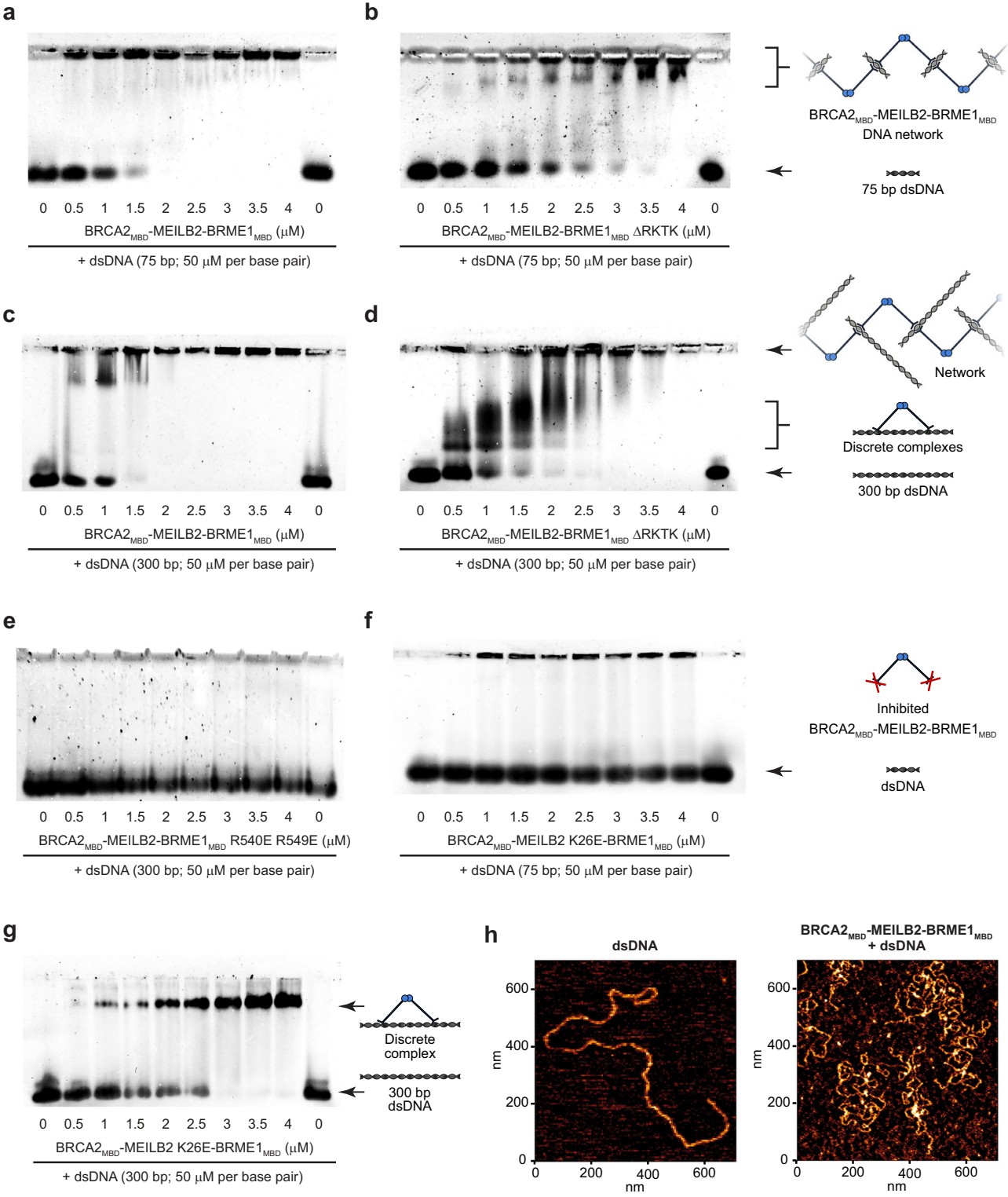

**Fig. 7 | BRCA2$_{MBD}$−MEILB2−BRME1$_{MBD}$ bridges between DNA molecules.** EMSAs analysing the ability of **a**, **c** BRCA2$_{MBD}$−MEILB2−BRME1$_{MBD}$ and **b**, **d** BRCA2$_{MBD}$−MEILB2−BRME1$_{MBD}$ ΔRKTK (truncation of BRME1 to remove the last four amino acids), at molecular concentrations indicated, to interact with **a**, **b** 75 base pair dsDNA and **c**, **d** 300 base pair dsDNA, at concentrations of 50 μM per base pair. EMSAs analysing the ability of BRCA2$_{MBD}$−MEILB2−BRME1$_{MBD}$ harbouring **e** BRME1 R540E R549E and **f**, **g** MEILB2 K26E mutations, at molecular concentrations indicated, to interact with **e**, **g** 300 base pair dsDNA and **f** 75 base pair dsDNA, at concentrations of 50 μM per base pair. **a**–**g** Representative of three independent experiments. **h** Atomic force microscopy images of dsDNA (linearised plasmid) in isolation (left) and upon binding by BRCA2$_{MBD}$−MEILB2−BRME1$_{MBD}$ complex (right); representative of three independent experiments. Source data are provided as a Source Data file.

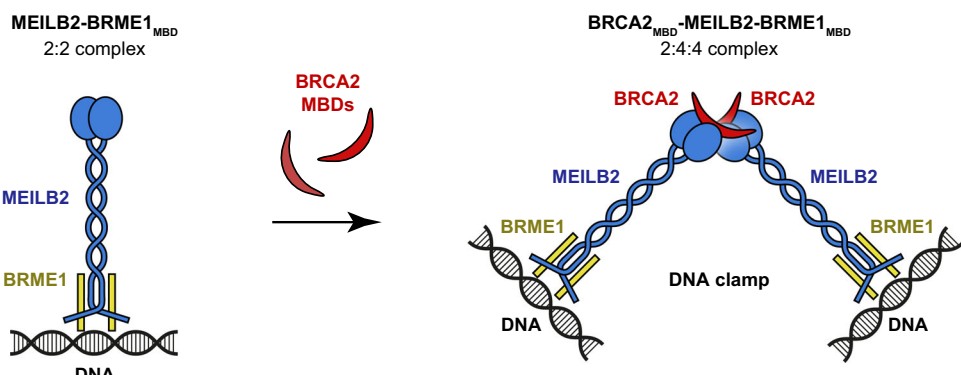

**Fig. 8 | Model of BRCA2$_{MBD}$-induced dimerisation of MEILB2–BRME1$_{MBD}$ into a V-shaped DNA clamp.** MEILB2–BRME1$_{MBD}$ 2:2 complexes bind to DNA via their N-terminal β-caps. BRCA2$_{MBD}$-binding induces dimerisation of MEILB2–BRME1$_{MBD}$ 2:2 complexes into a V-shaped 2:4:4 assembly, in which β-caps are physically separated by 25 nm and can clamp together DNA molecules. We propose that this could bridge between homologous chromosomes. Within the context of full-length molecules, a BRCA2-induced MEILB2–BRME1 DNA clamp may act in concert with the proximal DNA-binding domain and RAD51-/DMC1-binding regions of BRCA2, in a coordinated assembly to fulfil the specialised requirements of meiotic recombination.

## Discussion

The molecular programme of meiosis utilises the machinery of DNA DSB repair by homologous recombination to align and generate crossovers between homologous chromosomes that are critical for their correct segregation and the formation of haploid germ cells. The particular adaptations of meiotic recombination require several additional meiosis-specific components. These include the MEILB2–BRME1 complex, which binds directly to BRCA2 and is essential for the correct localisation of RAD51 and DMC1 recombinases to meiotic DSBs. Here, we combine crystallographic and cell biology data to report the structure of the MEILB2α–BRME1$_{MBD}$ 2:2 core complex, which we combine with existing structural data to generate an experimentally validated model of the BRCA2$_{MBD}$–MEILB2–BRME1$_{MBD}$ 2:4:4 ternary complex. This V-shaped structure contains DNA-binding sites at the tips of its two limbs, which are separated by 25 nm, suggesting that it may function to clamp together DNA molecules. As BRCA2$_{MBD}$-binding is necessary for dimerisation of MEILB2–BRME1$_{MBD}$ into the V-shaped structure, we conclude that its proposed role as a DNA clamp occurs within the context of its complex with BRCA2 (Fig. 8).

In this study, we analysed BRCA2–MEILB2–BRME1 complexes formed by the small MBDs of BRCA2 and BRME1. This was appropriate as the MBDs are located within unstructured regions of the proteins, so are unlikely to be part of wider structures. Further, MEILB2 binds to BRME1$_{MBD}$ and BRCA2$_{MBD}$ with low nano-molar affinity[44,51], and the consequent MEILB2–BRME1$_{MBD}$ 2:2 and BRCA2$_{MBD}$–MEILB2–BRME1$_{MBD}$ 2:4:4 complexes are highly stable and monodisperse in solution (this study). Thus, it is unlikely that other regions of BRME1 and BRCA2 interfere with these oligomers. Nevertheless, we cannot exclude this possibility as we currently lack experimental data regarding the full-length BRCA2–MEILB2–BRME1 complex. Hence, the following discussion assumes that the same structures and interactions occur within the context of the full-length proteins in vivo.

What is the function of the BRCA2-induced MEILB2–BRME1 DNA clamp in meiosis? We envisage two possibilities. First, it may bridge between DSB ends, in a manner that favours their repair by inter-homologue recombination. This is reminiscent of how MRN and CtIP are proposed to tether DNA ends and affect the choice of repair pathway at DSB sites[52–54]. However, it is difficult to see why this would be a specific requirement of meiotic recombination. Instead, it may bridge between homologous chromosomes to provide stabilising interactions between invading RAD51/DMC1-ssDNA nucleoprotein filaments and the template during inter-homologue recombination. This is our preferred model as it fulfils a unique requirement of meiosis, which is provided in mitotic recombination by cohesion between sister chromatids[55]. Notably, the 25 nm distance between DNA-binding sites of the dimerised MEILB2–BRME1 complex is comparable to the ~30 nm diameter of cohesin rings[56]. Hence, BRCA2-induced MEILB2–BRME1 DNA clamps may facilitate inter-homologue recombination by bridging between homologous chromosomes in the same manner as cohesin facilitates inter-sister recombination by tethering together sister chromatids. However, it is important to acknowledge that the functional significance of the DNA-binding activity of MEILB2–BRME1 has not yet been tested in vivo. Hence, the in vivo role (or even presence) of this DNA-binding activity is purely speculative and based solely on our in vitro biochemical findings.

A particular requirement of meiotic recombination is the removal of MEIOB2–SPATA22, in addition to RPA, from 3′-ssDNA overhangs for loading of RAD51 and DMC1 recombinases[16–18]. As BRCA2–DSS1 can offload RPA[12], it was proposed that MEILB2–BRME1 may displace MEIOB2–SPATA22 during meiotic recombination[22]. This is supported by the co-immunoprecipitation of MEIOB–SPATA22 with MEILB2, and the accumulation of MEIOB–SPATA22 at DSBs in MEILB2 and BRME1 knockouts[34,36,38]. However, it remains unknown whether MEILB2–BRME1 binds to MEIOB–SPATA22 directly. Our finding that MEILB2–BRME1$_{MBD}$ binds directly to DNA, with no overt specificity for single- or double-stranded substrates, is consistent with this proposed role. Indeed, as BRCA2's MBD is proximal to its globular DNA-binding domain, we speculate that BRCA2–DSS1 and MEILB2–BRME1 may act together, in a coordinated manner, to offload RPA and MEIOB–SPATA22 during meiotic recombination. This may occur alongside its proposed role in bridging between DNA ends or homologous chromosomes.

Thus far, we have addressed how BRCA2-binding affects MEILB2–BRME1 structure, but we must also consider the converse issue of how MEILB2–BRME1 affects the structure of BRCA2. In isolation, BRCA2 can form monomers, dimers and larger oligomers, in a manner that involves its N-terminal and C-terminal regions, and is modulated by ssDNA, DSS1 and RAD51[57,58]. Further, the BRCA2 dimer undergoes a conformational change upon binding to RAD51[11]. In meiosis, binding to MEILB2–BRME1 would clearly dimerise BRCA2 through its MBD region. However, given the lack of high-resolution structures of BRCA2 oligomers, it is not possible to say whether this would reinforce existing dimers or form alternative oligomers. In addition, binding to MEILB2–BRME1 may alter the conformation of BRCA2, such as to affect DMC1-binding and DNA binding of the sites immediately downstream of the MBD. Any effects on the oligomeric state and conformation of BRCA2 likely favour inter-homologue rather than inter-sister recombination[22]. Hence, these interactions are likely

to be deleterious if they occur outside of meiosis. Accordingly, MEILB2 is expressed in cancer cell lines and human tumour samples[35], and ectopic expression of MEILB2 inhibits homologous recombination in somatic cells[36,42]. Whilst BRME1 is also upregulated in cancers and suppresses recombination upon ectopic expression[36], it is not yet possible to speculate how it affects recombination as the structure and function of all but the MEILB2-binding site is unknown.

A recent parallel study reported the same MEILB2α–BRME1_MBD interface and BRCA2_MBD–MEILB2–BRME1_MBD 2:4:4 complex using human sequences[51], confirming that these structures are conserved between humans and mice. This study also reported that BRCA2_MBD–MEILB2 (in the absence of BRME1_MBD) forms 4:8 complexes that assemble into an interlocked ring structure[51]. In agreement with this, we observed that murine BRCA2_MBD–MEILB2 forms large structures in solution that are consistent with 4:8 oligomers (Supplementary Fig. 14). It remains unknown whether MEILB2 and BRME1 interact constitutively, or whether BRME1-binding dynamically controls the formation of interlocked rings or V-shaped clamp structures by BRCA2–MEILB2 during meiosis. It is possible that an interface within the interlocked ring structure may be responsible for the precipitation that we observed in solution, and may combine with bridging interactions to form the large-scale protein–DNA structures that remained unresolved during EMSA.

Overall, we propose that BRCA2-binding of MEILB2–BRME1 may clamp together homologous chromosomes, may facilitate the removal of MEIOB–SPATA22 from 3′-ssDNA overhangs, and may alter the oligomeric state and/or conformation of BRCA2. These are not mutually exclusive, so may represent multifactorial roles of MEIB2–BRME1 in meiotic recombination. Further, MEILB2–BRME1 likely acts in coordination with the DMC1-binding site and globular DNA-binding domain that are immediately downstream of the MBD. Hence, the ternary complex formed by MEILB2–BRME1, BRCA2–DSS1, RAD51 and DMC1 may operate as a single functional unit that fulfils the above roles to facilitate inter-homologue recombination. Elucidating the structure of this meiotic 'recombinosome', and its targeted disruption through mutagenesis in vivo, will ultimately reveal the molecular mechanisms that underpin meiotic recombination. Further, the unusual architecture imposed by MEILB2–BRME1 may provide unique structural insights that will uncover the molecular basis of the wider role of BRCA2 in recombination-mediated DNA repair.

## Methods

### Recombinant protein expression and purification

Constructs of mouse MEILB2 (amino-acid residues: 22–81, 22–338) and mouse BRCA2 (amino-acid residues: 2232–2283) were cloned into pMAT11[59] and pRSF-Duet1 (Merck Millipore) vectors for expression with an N-terminal TEV-cleavable His_6-MBP tag and an N-terminal TEV-cleavable MBP tag, respectively. Constructs of mouse BRME1 (amino-acid residues: 540–578 and 540–574) were cloned into pMAT11[59] and pRSF-Duet1 (Merck Millipore) vectors for expression with an N-terminal TEV-cleavable His_6 tag. Primer sequences used for cloning are included in the Source Data file. Protein constructs were expressed in BL21 (DE3) cells (Novagen), in 2xYT media, and induced with 0.5 mM IPTG for 16 h at 25 °C. Bacterial pellets were harvested, resuspended in 20 mM Tris, pH 8.0, 500 mM KCl, and lysed using a TS Cell Disruptor (Constant Systems) at 172 MPa. Cellular debris was later removed by centrifugation at $40,000 \times g$. Fusion proteins were purified through consecutive Ni-NTA (Qiagen), amylose (NEB), and HiTrap Q HP (Cytiva) ion exchange chromatography. The N-terminal tags were cleaved using TEV protease, and the cleaved samples were further purified through HiTrap Q HP (Cytiva) ion exchange chromatography and size-exclusion chromatography (HiLoad 16/600 Superdex 200, Cytiva) in 20 mM HEPES, pH 7.5, 150 mM KCl, 2 mM DTT for MEILB2–BRME1_MBD constructs and in 20 mM HEPES, pH 7.5, 325 mM KCl, 2 mM DTT for BRCA2_MBD–MEILB2–BRME1_MBD constructs. Purified protein samples

were spin-concentrated using Amicon Ultra centrifugal filter device (10,000 NMWL), flash frozen in liquid nitrogen, and stored at −80 °C. Purified proteins were analysed using SDS–PAGE and visualised with Coomassie staining (purified samples are shown in Supplementary Fig. 2). Protein concentrations were determined using Cary 60 UV spectrophotometer (Agilent) with extinction coefficients and molecular weights calculated by *ProtParam* (http://web.expasy.org/protparam/).

### Crystal structure of MEILB2α–BRME1_MBD core (PDB accession 7Z8Z)

MEILB2α–BRME1_MBD (22–81; 540–574) protein crystals grew upon incubation of protein at ~10 mg/ml on ice, in 20 mM HEPES pH 7.5, 150 mM KCl. Crystals were cryo-protected by the addition of 30% PEG 400, and were cryo-cooled in liquid nitrogen. X-ray diffraction data were collected at 0.9795 Å, 100 K, as four separate datasets, each of 3600 consecutive 0.10° frames of 0.020 s exposure on a Dectris Eiger2 XE 16 M detector at beamline I04 of the Diamond Light Source synchrotron facility (Oxfordshire, UK). Data were indexed, integrated in *XDS*[60], scaled and merged in *Aimless*[61], using *AutoPROC*[62]. Crystals belong to orthorhombic space group $P2_12_12_1$ (cell dimensions $a = 27.05$ Å, $b = 43.99$ Å, $c = 178.71$ Å, $\alpha = 90°$, $\beta = 90°$, $\gamma = 90°$), with one MEILB2–BRME1 2:2 complex in the asymmetric unit. Structure solution was achieved through fragment-based molecular replacement using *ARCIMBOLDO_LITE*[63], in which eight helices of 14 amino acids were placed by *PHASER*[64] and extended by tracing in *SHELXE* utilising its coiled-coil mode[45]. A correct solution was identified by a *SHELXE* correlation coefficient of 46.6%. Model building was performed through iterative re-building by *PHENIX Autobuild*[65] and manual building in *Coot*[66]. The structure was refined using *PHENIX refine*[65], using anisotropic (protein) and isotropic (water) atomic displacement parameters. The structure was refined against data to an isotropic resolution limit of 1.50 Å, to $R$ and $R_{\text{free}}$ values of 0.1965 and 0.2347, respectively, with 100% of residues within the favoured regions of the Ramachandran plot (0 outliers), clashscore of 1.85 and overall *MolProbity* score of 0.95[67].

### Size-exclusion chromatography multi-angle light scattering (SEC-MALS)

The absolute molar masses of protein samples were determined by multi-angle light scattering coupled with size-exclusion chromatography (SEC-MALS). Protein samples at >5 mg/ml were loaded onto a Superdex 200 Increase 10/300 GL size-exclusion chromatography column (Cytiva) in 20 mM HEPES, pH 7.5, 150 mM KCl, 2 mM DTT for MEILB2–BRME1 constructs and in 20 mM HEPES, pH 7.5, 500 mM KCl, 2 mM DTT for BRCA2–MEILB2–BRME1 constructs, at 0.5 ml/min, in line with a DAWN HELEOS II MALS detector (Wyatt Technology) and an Optilab T-rEX differential refractometer (Wyatt Technology). Differential refractive index and light scattering data were collected and analysed using *ASTRA* 6 software (Wyatt Technology). Molecular weights and estimated errors were calculated across eluted peaks by extrapolation from Zimm plots using a d$n$/d$c$ value of 0.1850 ml/g. Bovine serum albumin (Thermo Fisher Scientific) was used as the calibration standard.

### Size-exclusion chromatography small-angle X-ray scattering (SEC-SAXS)

SEC-SAXS experiments were performed at beamline B21 of the Diamond Light Source synchrotron facility (Oxfordshire, UK). Protein samples at concentrations >8 mg/ml were loaded onto a Superdex™ 200 Increase 10/300 GL size-exclusion chromatography column (GE Healthcare) in 20 mM HEPES pH 7.5, 150 mM KCl, 2 mM DTT for MEILB2–BRME1_MBD constructs and in 20 mM HEPES pH 7.5, 500 mM KCl, 2 mM DTT for BRCA2_MBD–MEILB2–BRME1_MBD constructs at 0.5 ml/min using an Agilent 1200 HPLC system. The column outlet was fed into the experimental cell, and SAXS data were recorded at

12.4 keV, detector distance 4.014 m, in 3.0 s frames. *ScÅtter* 3.0 was used to subtract, average the frames and carry out the Guinier analysis for the radius of gyration (*Rg*), and *P*(*r*) distributions were fitted using *PRIMUS*[68]. Crystal structures and models were fitted to experimental data using *CRYSOL*[69].

#### DNA-induced precipitation

To the purified MEILB2–BRME1$_{MBD}$ wildtype and BRME1 R540E R549E complexes (30 μM), 75 bp dsDNA (the same substrate used for EMSA) was gently mixed to a final concentration of 1 μM (per molecule) in a buffer containing 20 mM HEPES (pH 7.5), 150 mM KCl. The samples were then centrifuged at high speed (17,000 × *g*) for 30 min. The pellets and supernatant solutions were analysed on SDS–PAGE.

#### Protein–DNA binding by streptavidin pulldown

The purified MEILB2–BRME1$_{MBD}$ complex (30 μM) was incubated with and without 1 μM (per molecule) of 75 bp dsDNA (the same substrate used for EMSA) labelled with 3′ Biotin-TEG (Integrated DNA Technologies), as well as with 30 μM of MEILB2–BRME1$_{MBD}$ R540E R549E in the presence of 1 μM of 75 bp DNA labelled with 3′ Biotin-TEG, in a buffer containing 20 mM HEPES (pH 7.5), 150 mM KCl, and 10% glycerol. These mixtures were incubated for 15 min at room temperature. Pierce Streptavidin Magnetic Beads (Thermo Scientific; 30 μL/reaction) were first washed three times with 1 mL of a high-salt buffer (20 mM HEPES, 1 M KCl), then three times with a low-salt buffer (20 mM HEPES, 150 mM KCl, 10% glycerol). The samples were incubated with the beads for 10–15 min, after which unbound material was removed by discarding the supernatant following magnetic separation. The beads were then washed five times with 1 mL of the low-salt buffer, and elution was performed using 20 mM HEPES (pH 7.5), 100 mM KCl, 50 μM biotin, and 1 unit of Benzonase Nuclease. The eluted samples were analysed using SDS–PAGE.

#### Electrophoretic mobility shift assays (EMSAs)

Protein complexes were incubated with 50 μM (per base pair) 75 bp and 300 bp linear random sequence dsDNA, 75 base FAM-poly(dT) ssDNA and 75 base/bp ssDNA–dsDNA junction, at concentrations indicated, in 20 mM HEPES pH 7.5, 325 mM KCl for 60 min at room temperature. Glycerol was added at a final concentration of 3%, and samples were analysed by electrophoresis on a 0.4% (w/v) agarose gel in 0.5x TBE pH 8.0 at 20–40 V for ~4 h at 4 °C. DNA was detected by SYBR™ Gold (Thermo Fisher). The sequences of the DNA substrates are included below.

**75 bp dsDNA**. ATGGAATCTAAAGAAGAATTTGTTAAAGTCAGAA AGAAGGACCTGGAACGGCTGACGACGGAAGTGATGCAAATA.

**300 bp dsDNA**. ATGGAATCTAAAGAAGAATTTGTTAAAGTCAGAA AGAAGGACCTGGAACGGCTGACGACGGAAGTGATGCAAATACGGGAC TTCTTACCCAGAATACTAAATGGGGAGTTACTGGAAAGTTTCCAGA AATTAAAGATGGTAGAAAAAAACCTGGAAAGAAAAGAACAAGAATT AGAGCAACTGATAATGGACCGTGAACACTTCAAAGCCCGGCTAGAAA CTGCACAGGCAGACAGTGGGAGGGAGAAGAAGGAGAAGTTGGCTC TTCGACAGCAGCTGAATGAGGCAAAACAGCAGCTC.

**75 base FAM-poly(dT) ssDNA**. FAM-TTTTTTTTTTTTTTTTTTTT TTTTTTTTTTTTTTTTTTTTTTTTTTTTTTTTTTTTTTTTTTTTTTTTTTT TTT.

**75 base/bp ssDNA–dsDNA junction**. CCGGCCCGCGGTGTCCGCG GGCCGGTTTTTTTTTTTTTTTTTTTTTTTTTTTTTTTTTTTTTTTTTTTT TTTTTT.

#### $K_D$ determination by EMSA

Quantification of DNA binding was performed through EMSA (as described above) using 50 nM FAM-labelled 75 bp random sequence dsDNA and 100 nM 75 nt poly(dT), at protein concentrations indicated. DNA was detected by FAM and SYBR™ Gold (Thermo Fisher) staining using a ChemiDoc MP Imaging System (Bio-Rad). Gels were analysed using Image Lab software (Bio-Rad). The DNA-bound proportion was plotted against molecular protein concentration and fitted to the Hill–Langmuir equation (below), with apparent $K_D$ and Hill coefficient (*n*) determined, using Prism8 (GraphPad). Protein concentrations used for $K_D$ estimation are quoted for the oligomeric species.

$$\% \ bound \ DNA = \frac{C^n}{K_D{}^n + C^n} \tag{1}$$

#### Atomic force microscopy

The purified BRCA2$_{MBD}$–MEILB2–BRME1$_{MBD}$ complex was diluted to a concentration of 1–3 ng/μL in an imaging buffer containing 20 mM HEPES (pH 7.5), 325 mM KCl, with 0.6 ng/μL of linear double-stranded DNA. The protein–DNA mixture was then incubated at room temperature for 10 min. To facilitate DNA adsorption onto mica, 10 μL of filtered 0.001% poly-L-lysine solution was incubated on freshly cleaved mica for 20 s. The mica surface was then washed three times with 1 mL of filtered Bio Performance Certified (BPC) water and dried under a stream of filtered nitrogen gas. Finally, 10 μL of the protein–DNA mixture was adsorbed onto the mica surface for 10 min. Unbound material was subsequently washed off with the imaging buffer, followed by washes with BPC water, and finally dried under a stream of filtered nitrogen gas. Imaging was performed in air at room temperature, using a FastScan-A probe in Tapping on a FastScan Bio Atomic Force Microscope (Bruker).

#### Structural modelling of the full MEILB2–BRME1$_{MBD}$ 2:2 complex

Models were generated using a local installation of *AlphaFold2* v2.3.2[50] that was modified to control the use of PDB structures and newly solved crystal structures as templates. Models of the full MEILB2–BRME1$_{MBD}$ 2:2 complex were generated through the multimer pipeline[49], using the MEILB2α–BRME1$_{MBD}$ 2:2 core crystal structure 7Z8Z reported herein, and PDB structure 7LDG[43], as the sole templates. Modelling data were analysed using modules from the ColabFold notebook[70]. A deformed version of the MEILB2–BRME1$_{MBD}$ model was generated by introducing a turn at the kink between α1 and α2 helices of MEILB2, such that α1 helices and BRME1$_{MBD}$ chains pack against α2 helices. This was performed using *Coot*[66] and *PyMOL* Molecular Graphics System, Version 2.4 Schrödinger, LLC.

#### Structural modelling of the BRCA2–MEILB2–BRME1 2:4:4 complex

A model of the BRCA2$_{MBD}$–MEILB2–BRME1$_{MBD}$ 2:4:4 complex was generated by docking two MEILB2–BRME1$_{MBD}$ 2:2 complex models onto opposing MEILB2 ARM domain dimers of the BRCA2$_{MBD}$–MEILB2 ARM 2:4 crystal structure (PDB accession 7LDG)[43]. The docked MEILB2–BRME1$_{MBD}$ 2:2 complexes and BRCA2$_{MBD}$ chains from the structure were then combined. Models were generated using *PyMOL* Molecular Graphics System, Version 2.4 Schrödinger, LLC.

#### Cryo-electron microscopy (cryo-EM)

R1.2/1.3 300-mesh copper grids (Quantifoil) were glow discharged for 90 s prior to sample freezing. A volume of 4 μl of purified BRCA2$_{MBD}$–MEILB2–BRME1$_{MBD}$ complex (1 mg/ml) was dispensed onto the grid, blotted for 2 s and flash frozen in liquid ethane cooled with liquid nitrogen using a Vitrobot Mark IV (Thermo Fisher) operated at 4 °C and 100% humidity. Data were collected on an FEI Tecnai F20 cryo-electron microscope operated at 200 kV equipped with a K2 Summit direct electron detector (Gatan Inc.), using *SerialEM*[71]. Movies were recorded in electron-counting mode fractionating with a total exposure of 39 e/Å². A defocus range of −0.8 to −1.5 μm was used and

the physical pixel size was 1.02 Å/pixel. The movies were gain normalised, motion-corrected, and dose-weighted with *MotionCo2*[72]. *CryoSPARC* v4.3.1[73] was used to import micrographs, perform CTF estimation with *CTFFIND4*[74], pick particles, and to generate 2D class averages and a 3D ab initio model.

### Protein sequence and structure analysis
MEILB2 and BRME1 sequences were aligned and visualised using MUSCLE[75] and Jalview[76]. Molecular structure images were generated using the PyMOL Molecular Graphics System, Version 2.4 Schrödinger, LLC.

### Mice
Wild-type mice were congenic with the C57BL/6J background. The mice are housed in IVC cages with a 12 h dark and light cycle. The temperature is 20–22 °C and the relative humidity is between 45 and 60%. The mice have bedding material in the form of wood shavings and wood litter as well as a house of paper and nesting pads as enrichment. Cage changing is done at least once a week. All animal experiments were approved by the Regional Ethics Committee of Gothenburg, governed by the Swedish Board of Agriculture (#1316/18).

### Antibodies
The following antibodies were used: rabbit antibodies against GFP (Invitrogen; A11122, 2339829, 1:1000 for WB and 1:500 for IF); mouse antibodies against β-Actin (Sigma; A2228-200UL, 067M4856V, 1:2000 for WB); and chicken antibody against SYCP3 (Shibuya lab, 1:5000 for IF).

### Exogenous protein expression in the testis
Plasmid DNA was electroporated into live mouse testes as previously described[77]. Briefly, male mice at postnatal days 16–20 were anaesthetised, and the testes were pulled from the abdominal cavity. Plasmid DNA (10 μl of 5 μg/μl solution) was injected into each testis using glass capillaries under a stereomicroscope. Testes were held between a pair of tweezers-type electrodes (CUY21; BEX), and electric pulses were applied four times and again four times in the reverse direction at 35 V for 50 ms for each pulse. The testes were then returned to the abdominal cavity, and the abdominal wall and skin were closed with sutures. The testes were removed 24 h after the electroporation, and immunostaining was performed.

### Immunostaining of spermatocytes
Testis cell suspensions were prepared by mincing the tissue with flathead forceps in PBS, washing several times in PBS, and resuspending in a hypotonic buffer (30 mM Tris (pH 7.5), 17 mM trisodium citrate, 5 mM EDTA, 2.5 mM DTT, 0.5 mM PMSF, and 50 mM sucrose). After 30 min, the sample was centrifuged and the supernatant was aspirated. The pellet was resuspended in 100 mM sucrose. After 10 min, an equal volume of fixation buffer (1% paraformaldehyde and 0.1% Triton X-100) was added. Cells were applied to a glass slide, allowed to fix for 2 h at room temperature, and air-dried. For immunostaining, the slides were incubated with primary antibodies in PBS containing 5% BSA for 2 h and then with the following secondary antibodies for 1 h at room temperature: Donkey Anti-Rabbit Alexa 488 (1:1000; Invitrogen; A21206, 2376850) and Donkey Anti-Chicken Alexa 594 (1:1000; Invitrogen; A78951, 2551396). The slides were washed with PBS and mounted with VECTASHIELD medium with DAPI (Vector Laboratories).

### Microscopy
Images were obtained on a microscope (Olympus IL-X71 Delta Vision; Applied Precision) equipped with 100× NA 1.40 objective, a camera (CoolSNAP HQ; Photometrics), and *softWoRx* 5.5.5 acquisition software (Delta Vision). Images were processed with Photoshop (Adobe).

### Reporting summary
Further information on research design is available in the Nature Portfolio Reporting Summary linked to this article.

## Data availability
Crystallographic structure factors and atomic coordinates have been deposited in the Protein Data Bank (PDB) under accession number 7Z8Z, and raw diffraction data have been uploaded to https://proteindiffraction.org/. This study used PDB entries 7LDG and 7BDX. All data supporting the findings of this study, in addition to raw gel images, are provided in the accompanying Supplementary Information and Source data files. Source data are provided with this paper.

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

## Acknowledgements

We thank Diamond Light Source and the staff of beamlines I04 and B21 (proposal mx25233), the cryo-EM facility in the School of Biological Sciences at the University of Edinburgh (SULSA and Wellcome Grant 087658/Z/08/Z) and the Atomic Force Microscopy Facility in the School of Engineering at the University of Edinburgh. This work was supported by a Wellcome Senior Research Fellowship 219413/Z/19/Z (O.R.D.), a core grant to the Wellcome Centre for Cell Biology 203149 (O.R.D.), the Structural Biology core of the Wellcome Discovery Research Platform for Hidden Cell Biology 226791 (O.R.D.), and the European Research Council StG-801659 (H.S.).

## Author contributions

M.G. performed all the in vitro experimental work; J.Z. and K.Z. performed the in vivo experimental work under the supervision of H.S.; M.G. and O.R.D. designed experiments and analysed data; and O.R.D. wrote the manuscript.

## Competing interests

The authors declare no competing interests.
