## [Peer Review File · Nature Communications]

MEILB2-BRME1 forms a V-shaped DNA clamp upon BRCA2-binding in meiotic recombinationREVIEWER COMMENTS

Reviewer #1 (Remarks to the Author):

Gurusaran et al. present a structural and biochemical characterisation of a complex of BRME1 with MEILB2. The authors use in vivo approaches to show that the formation of the MEILB2 BRME1 complex is necessary to facilitate its recruitment to DNA break sites in meiosis. Analysis of the N-terminal end of the BRME1-MEILB2 complex suggested that it might bind to DNA, which was partially confirmed using EMSAs. The MEILB2-BRME1 complex is further dimerised by the previously characterised MEILB2 binding domain of BRCA2 to form a large 2:4:4 complex. The 2:4:4 complex likely forms a V-shaped structure, with the DNA binding domains oriented some 25 nm away from another. The authors make some interesting proposal about what the functionality of this complex might be in meiotic recombination.

This is an important study that significantly furthers our understanding of how DNA repair takes place in meiosis. The paper is very well written, clearly structured, and the arguments made are mostly supported by the data. Likewise the data is clearly of a very high quality, in particular the use of SEC-SAXS coupled with experimental and predicted structures is a powerful approach. I am very supportive of publication, and think it would be suitable and interesting for a wide ranging audience. The only area that needs to be strengthened is regarding the DNA binding. While the authors have made judicious use of mutants, there is still the risk that EMSAs can give rise to some artefacts in nucleic acid binding. Given the importance placed on the DNA binding activity of the complex both in the title, abstract and discussion, this does need some work. It would be necessary to show DNA binding through at least one other method, as described in detail below. My remaining "major points" can largely be addressed through re-analysis of existing data.

Major points

The beta-cap is an interesting structure. Could the authors comment on how well conserved this is, perhaps in relation to the rest of MEILB2? Also, if there is a motif associated with the beta-cap, could the authors comment on whether this appears elsewhere, perhaps among other DNA binding/repair proteins?

Structure of the MEILB2-BRME1 complex. While the AlphaFold2 model of the full 2:2 complex looks very reasonable, it might be worth determining the chi-squared values of some alternative shapes. How, for example, might it look if it were a more compact structure? I appreciate that the authors have not done this yet, since it seems like they would be straw man structures to be easily invalidated by the SAXS data. However, here I think it would strengthen the authors' argument, especially for an audience unfamiliar with SAXS. Related to this, if the authors remove the constraints on the AlphaFold model generation (imposed by a limited template choice) is there a greater variety of models?

MEILB2-BRME1 binds to DNA via its beta-cap. I would really like to see some additional DNA binding experiments. Ideally an additional method, whether FP, IST or SPR. It could also be a qualitative method using, for example, biotinylated DNA under conditions with a reasonable salt concentration. Can the DNA-protein complex form in a size exclusion experiment? If a SEC stable complex can be produced, some additional characterisation would also be very insightful (at least SEC-MALS, ideally SEC-SAXS). While the Hill equation was used in Figure 5d, I couldn't find the value of the Hill Coefficient used.

BRCA2 dimerises MEILB2-BRME1 to form a V-shaped assembly. Here it would be nice to show a SDS-PAGE gel for Figure 6a (or a gel from a similar experiment). I ask for this because it would be helpful to show that the MEILB2-BRME1 complex doesn't undergo (an albeit unlikely) major structural rearrangement that could account for a different interpretation of the SEC-MALS data. Could the authors show the molecular mass of a complex formed from the addition of a larger BRCA2 construct (e.g. MBP-BRCA2). Additionally, how well does the 161 kDa measured value match the theoretical size of the 2:4:4 complex? Figure 6c needs a scale bar, and possibly some additional annotation to help orientate the reader.

BRCA2-MEILB2-BRME1 acts as a DNA clamp. As above, the authors should show some additional characterisation of DNA binding using other approaches. In addition a useful control here would be what happens if one uses a mutant BRCA2 that cannot bind to the MEILB2-BRME1 complex? For the complexes that don't enter the gel, the authors should also acknowledge that these could be protein-DNA aggregates, rather than an organised network. The authors also suggest that the two beta-caps of the ternary complex interact cooperatively with a long DNA molecule. If this is true, the authors should quantitate their EMSAs and compare the Hill Coefficients used for a BRCA2-MEILB2 K26E BRME1 ternary complex on DNA versus one with a mutation that prevents BRCA2 mediated complex dimerisation. Furthermore, if the ternary complex binds DNA more tightly than the MEILB2-BRME1 complex, then might this not be SEC stable?

Minor points

Page 11, first paragraph. Figure 6a rather than Figure 5a

Figure 5c, odd looking box around gel label

Figure 5d, please use either nt or bp for the DNA length.

Page 12, second paragraph. Wouldn't 75 bp B-form DNA have an approximate length of 25.5 nm rather than 22 nm? Assuming 3.4 Å rise per bp.

Supplementary Figure 4b - please somehow make the colouring of the structures clear in the figure (e.g. make the label PDB 7LDG red)

Reviewer #2 (Remarks to the Author):

BRCA2 is a major player in the control of HR and a few years ago several studies identified two proteins MEILB2 (also known as HSF2BP) and BRME1 (aka MEIOK21, MMER, C19ORF57 and 2930432K21Rik) as directly involved in BRCA2 activity. They interact with BRCA2 and somehow promote its loading on DNA. Previous studies have identified interactions between MEILB2 and BRCA2 and between MEILB2 and BRME1 and determined the structure of the BRCA2-MEILB2 complex (using the interacting protein domains). Further characterization of these proteins and their interactions is essential to understand how the ssDNA tails become accessible, and then loaded by the recombinases RAD51 and DMC1., and the challenge is to integrate the different partners involved.

Here the authors gain insight into this direction by determining the structure of the MEILB2-BRME1 complex, then by adding the BRCA2 binding domain, they show a tetramerization of the complex including domains from all three proteins (BRCA2/BRME1/MEILB2). One of the new findings is that the MEILB2-BRME1 complex binds DNA, and the authors propose a function in bridging DNA helices when the complex is tetrameric. All these data are important, and necessary to gain insight into these proteins. As these proteins are truncated in several assays, it is difficult to determine what the in vivo behaviour will be. The BRCA2/BRME1/MEILB2 complex contains only a small domain of both BRME1 and BRCA2. This is a problem because BRCA2 has other domains with affinity for the DNA, which may strongly influence the overall properties of this complex. The author's interpretation is highly speculative and should therefore take this into account, as alternative scenarios are possible.

Of note, a similar study has been very recently published: Ghouil, R. et al. (2023). BRCA2-HSF2BP oligomeric ring disassembly by BRME1 promotes homologous recombination. *Sci Adv* 9, eadi7352. 10.1126/sciadv.adi7352. In this study, the interactions are tested in different ways (for instance MEILB2 with BRCA2 but without BRME1, human proteins) thus leading to different observations. The authors should discuss one of the common observations, which is the tetrameric form of the BRCA2/BRME1/MEILB2 complex. It would be interesting to know if they have also observed the oligomeric form in the absence of BRME1.

A general comment on the presentation of the data is that often one does not know which protein domain is being tested, and these protein domains are inappropriately named as if they were full-length proteins, which is misleading. Thus, one has to search for methods or various parts of the text to get the information. This should be revised through the text, figures and legends.

Other comments

1) MEILB2-BRME1: Please check whether the domain is 22-81 or 22-79. Also, a comment is needed to understand why the first 21 aa have been removed (probably solubility issues), and what would be their predicted structure. This could be important information given the DNA interactions proposed in this study involving residues at the N ter of MEILB2.

2) Fig3b and Sup Fig3: which fraction of the samples are loaded on the gels?

3) Is the BRME1 3E mutant protein properly folded in vitro ?

4) DNA binding assays: which protein domains are used in the assays.

Since the in vivo substrate is a dsDNA with ssDNA tails, the authors should test whether there is a specific affinity for ssDNA-dsDNA junctions. What is the so-called "random sequence DNA" of 75bp and 300bp?

Do the mutations introduced (on K26 and R540-549) have any effect on the dimerization? The authors conclude that the DNA binding is a consequence of the complex formation. To conclude this, data about the DNA binding affinity of MEILB2 and BRME1 separately should be shown.

5) Complex with BRCA2. The proposition of a V-shaped complex is potentially interesting however not supported by the data: on the CryoEM (fig6c), angles are quite variable. So, the claim that the angle is approximately 90° is not convincing. Quantitative measurements of the angle and the distance between the ends of the arms should be performed. Is the

tetramer formed in the presence of K26E?

The model of bridging the two ends is not convincing. The first point is that there is no need to bridge the two ends for repair by HR, on the contrary, current models propose that the two ends act differently, one searching for the homolog, and the other potentially interacting with the sister chromatin. The only scenario where one does expect a bridging of two ends is in the NHEJ pathways, which is not the case here. Bridging homologs or sister chromatids may be more relevant.

Others

P3: typo: 3' single-stranded DNA

P4: binds together homologous chromosome axes.

P11: Figure 5a: should be 6a

BRME1 RGTK deletion: does it affect MEILB2 binding?

Reviewer #3 (Remarks to the Author):

In this manuscript, Gurusaran et al. studied the structure of MEILB2-BRME1 complex with or without BRCA2 binding. They found that MEILB2-BRME1 complex binds both ssDNA and dsDNA with its N-terminal-cap. The ARM domain of MEILB2, which locates on its C-termini, binds to the MBD domain of BRCA2 and dimerize to form a V-shaped DNA clamp. With these findings, they proposed that the BRCA2-MEILB2-BRME1 complex may function as a DNA clamp to connect the resected DNA ends or homologous chromosomes to facilitate meiotic recombination. The results and structure analysis of MEILB2-BRME1 complex are reliable and of certain significance. However, several points need to be concerned:

Major concerns:

1. The main problem for this research is that all the conclusions and proposals are based on in vitro results. The proposed structure of BRCA2-MEILB2-BRME1 complex could not answer how it facilitates RAD51 and DMC1 loading. Also, there is no evidence to show whether the proposed function of BRCA2-MEILB2-BRME1 based on its V-shaped DNA clamp structure exists in meiosis or not. In fact, *Brca2* Δ 12/ Δ 12 mice, whose BRCA2 does not interact with MEILB2, showed no defects in HR, indicating that the proposed function of BRCA2-MEILB2-BRME1 based on its V-shaped DNA clamp structure is dispensable in meiosis.
2. Though MEILB2-BRME1 could bind DNA directly, their recruitment requires MEIOB and SPATA22, indicating that these proteins may interact with each other when they are recruited. Thus, the presence of MEIOB and SPATA22 could also change the structure of BRCA2-MEILB2-BRME1 complex in vivo, which should be considered when analyzing the structure and function of BRCA2-MEILB2-BRME1 complex.
3. BRCA2-MEILB2-BRME1 formed a 2:4:4 complex when BRCA2-MBD is added. However, BRCA2 is a large protein that contained 3148aa rather than the 52aa BRCA2-MBD used in this research. Could it be possible that BRCA2-MEILB2-BRME1 complex forms 1:2:2 rather than 2:4:4 structure at the presence of full length BRCA2?
4. It seemed that MEILB2-BRME1 binds both ssDNA and dsDNA. Given that the known function of BRCA2-MEILB2-BRME1 is to facilitate the replacement of RPA by RAD51 and DMC1, which takes place on ssDNA, the authors should pay more attention on the interaction of BRCA2-MEILB2-BRME1 with ssDNA or RPA coated ssDNA, rather than dsDNA.

Taken together, the significance of this research is limited.

Response to reviewers' comments
'MEILB2-BRME1 forms a V-shaped DNA clamp upon BRCA2-binding in
meiotic recombination'
(NCOMMS-23-48925-T)

We are pleased that the reviewers recognise the value of this work, and are grateful for their thoughtful and considered comments. Here, we provide responses to the questions and comments raised by the reviewers, and outline how, in accordance with their requests, we have revised the manuscript.

Reviewer #1 (Remarks to the Author):

Gurusaran et al. present a structural and biochemical characterisation of a complex of BRME1 with MEILB2. The authors use in vivo approaches to show that the formation of the MEILB2 BRME1 complex is necessary to facilitate its recruitment to DNA break sites in meiosis. Analysis of the N-terminal end of the BRME1-MEILB2 complex suggested that it might bind to DNA, which was partially confirmed using EMSAs. The MEILB2-BRME1 complex is further dimerised by the previously characterised MEILB2 binding domain of BRCA2 to form a large 2:4:4 complex. The 2:4:4 complex likely forms a V-shaped structure, with the DNA binding domains oriented some 25 nm away from another. The authors make some interesting proposal about what the functionality of this complex might be in meiotic recombination.

This is an important study that significantly furthers our understanding of how DNA repair takes place in meiosis. The paper is very well written, clearly structured, and the arguments made are mostly supported by the data. Likewise the data is clearly of a very high quality, in particular the use of SEC-SAXS coupled with experimental and predicted structures is a powerful approach. I am very supportive of publication, and think it would be suitable and interesting for a wide ranging audience. The only area that needs to be strengthened is regarding the DNA binding. While the authors have made judicious use of mutants, there is still the risk that EMSAs can give rise to some artefacts in nucleic acid binding. Given the importance placed on the DNA binding activity of the complex both in the title, abstract and discussion, this does need some work. It would be necessary to show DNA binding through at least one other method, as described in detail below. My remaining "major points" can largely be addressed through re-analysis of existing data.

Major points

The beta-cap is an interesting structure. Could the authors comment on how well conserved this is, perhaps in relation to the rest of MEILB2? Also, if there is a motif associated with the beta-cap, could the authors comment on whether this appears elsewhere, perhaps among other DNA binding/repair proteins?

1. The β -cap shows a high level of conservation across animals, comparable with that of the wider $\alpha 1$ region and ARM domain. In contrast, the $\alpha 2$ region (which largely acts as a coiled-coil spacer) is more divergent, and the N-terminal tail is very poorly conserved. This conservation pattern is consistent with the β -cap having an important function in the same manner as $\alpha 1$ and ARM regions in BRME1- and BRCA2-binding, respectively. We have added a comment about this in the results section (line numbers 159-160) and have included a full alignment across a selection of animal species in Supplementary Figure 1 so readers can view the conservation. In this, we have indicated the locations of the β -cap, $\alpha 1$, $\alpha 2$ and ARM regions.

The β -cap is very short (seven amino-acids), and the only clear pattern in the sequence is an initial aromatic amino-acid (F24) followed by four alternating hydrophobic and hydrophilic residues that form the β -strand (25-VKVR-28). Its structure is likely highly dependent on context given its interaction with downstream amino-acids of MEILB2 and bound BRME1 sequence. Hence, we do not think that it is possible to define a motif that would differentiate between the β -cap and any other β -strand. We cannot identify similar β -cap structures in any other DNA-binding/repair or other proteins. The closest we have found is the presence of β -layers as interruptions within coiled-coils (e.g. PDB accession 2BA2; discussed in https://doi.org/10.1007/978-3-319-49674-0_3). However, we cannot find any cases in which the β -layer caps off the coiled-coil in the same manner as we observe for MEILB2-BRME1. We have extended our discussion of this in the results section (line numbers 160-165).

Structure of the MEILB2-BRME1 complex. While the AlphaFold2 model of the full 2:2 complex looks very reasonable, it might be worth determining the chi-squared values of some alternative shapes. How, for example, might it look if it were a more compact structure? I appreciate that the authors have not done this yet, since it seems like they would be straw man structures to be easily invalidated by the SAXS data. However, here I think it would strengthen the authors' argument, especially for an audience unfamiliar with SAXS. Related to this, if the authors remove the constraints on the AlphaFold model generation (imposed by a limited template choice) is there a greater variety of models?

2. We agree that it is beneficial to show the chi-squared value of an alternative model. As the reviewer outlines, there is a danger of using a model that is so different that it is bound to conflict with the SAXS data. Hence, we have built a deformed model by introducing a turn at the kink between $\alpha 1$ and $\alpha 2$ helices of MEILB2, such that $\alpha 1$ helices and BRME1 chains are packed against $\alpha 2$ helices. This deformed model is rather similar to the true structure except that it is shorter (13 nm rather than 18 nm). This deformed model fitted to the experimental data with much larger residuals (especially in the low-q range – in keeping with the difference in molecular length) and a chi-squared value of 21.08 (in comparison with 2.96 for the original 2:2 model). These additional data demonstrate that SAXS analysis is sensitive to relatively small structural

changes, and provides additional confidence that the presented full 2:2 model is correct. We have added these additional SAXS data to modified Figure 4b, have shown the deformed model in Supplementary Figure 6d, and have added additional description in the results section (line numbers 220-225).

AlphaFold2 correctly builds the overall full 2:2 structure whether or not we impose the use of templates. However, whilst the ARM domain region is always very accurate, the MEILB2-BRME1 core region is often slightly distorted if we do not include its structure (7Z8Z) as a template. The choice to impose templates was to ensure that the modelled structure includes MEILB2-BRME1 core and ARM domain regions that closely match their solved crystal structures, essentially using AlphaFold2 to join these two structures together by building the short intervening coiled-coil. It was not intended to compare the performance of AlphaFold2 with and without these templates.

MEILB2-BRME1 binds to DNA via its beta-cap. I would really like to see some additional DNA binding experiments. Ideally an additional method, whether FP, IST or SPR. It could also be a qualitative method using, for example, biotinylated DNA under conditions with a reasonable salt concentration. Can the DNA-protein complex form in a size exclusion experiment? If a SEC stable complex can be produced, some additional characterisation would also be very insightful (at least SEC-MALS, ideally SEC-SAXS). While the Hill equation was used in Figure 5d, I couldn't find the value of the Hill Coefficient used.

3. We agree with the reviewer that it would be ideal to test DNA-binding of the MEILB2-BRME1 complex by additional approaches. However, this has proven problematic owing to the technical restriction that DNA induces the precipitation of the MEILB2-BRME1 complex when added at a physiological salt concentration of 150 mM KCl. Importantly, this DNA-induced precipitation is abrogated by the BRME1 R540E R549E mutation. This effect is illustrated in the image below, which shows the appearance of the MEILB2-BRME1 2:2 complex at 150 mM KCl, upon addition of DNA, and for the mutant upon addition of DNA.

Condition: 20mM HEPES+150mM KCl

1. Meilb2-Brme1 (30uM)
2. Meilb2-Brme1 (30uM) + DNA (1uM)
3. Meilb2-Brme1 R540E R549E (30uM) + DNA (1uM)

This DNA-induced precipitation means that it is not possible to study DNA-binding in solution by methods such as FP, ITC or SEC at physiological salt concentrations. Further, at a higher salt concentration (325 mM KCl), which prevented precipitation, the complex dissociated on SEC. Instead, we found that EMSAs provided a suitable means for visualising these complexes. In these experiments, we set up reactions at 325 mM KCl, and then relied on reduction in salt concentration as the complex entered the gel and transferred into the TBE running buffer. This appeared to stabilise complexes without forming large assemblies/aggregates (other than at high concentrations), presumably owing to individual complexes being physically separated before they could aggregate. Hence, this is the reason for the use of EMSAs throughout the manuscript. We have added a discussion of the issue of DNA-induced precipitation in the manuscript (line numbers 233-246) and have included an SDS-PAGE of the supernatant and pellet of the above experiment in Figure 5b.

Nevertheless, to alleviate the reviewers' concerns, we sought to find some additional methods in which we could validate the DNA-binding findings that we observed by EMSA. In keeping with the reviewers' suggestions, we were able to demonstrate binding using a biotinylated DNA substrate and streptavidin magnetic beads at a physiological salt concentration of 150 mM KCl. This experiment nicely confirmed DNA-binding by the MEILB2-BRME1 complex, which was abrogated by the BRME1 R540E R549E mutation. These data have been added to the manuscript as Figure 5c (line numbers 235-237).

In addition, we used atomic force microscopy to visualize directly the structures formed by the BRCA2-MEILB2-BRME1 complex on DNA. Here, we set up reactions at 325 mM KCl, adsorbed on mica and then washed in a physiological salt concentration of 150 mM KCl. In these experiments, BRCA2-MEILB2-BRME1 formed large structures on DNA molecules that appeared to establish links between or within molecules, resulting in DNA molecules becoming heavily tangled and looped. These structures are precisely what we would expect given our model that BRCA2-MEILB2-BRME1 tethers together DNA molecules. Importantly, we did not observe tangled or looped DNA when using BRCA2-MEILB2-BRME1 harbouring the BRME1 R540E R549E mutation. We have included these data in the manuscript as Figure 7h and Supplementary Figure 13 (line numbers 359-367).

Together, we believe that the streptavidin magnetic bead data for the MEILB2-BRME1 complex, the atomic force microscopy data for the BRCA2-MEILB2-BRME1 complex, and the mutant data in both cases, provide suitable confirmation for the DNA-binding reported by EMSA in this manuscript.

The reviewer asks about the Hill coefficient used in Figure 5d (now Figure 5f). We fitted both the K_d and Hill coefficient to the data, and the fitted Hill coefficients (n) are shown in the panel. These are 3 and 2 for dsDNA and ssDNA, respectively, indicating

positive cooperativity. We have clarified that n is the Hill coefficient in the figure legend (line number 1010).

BRCA2 dimerises MEILB2-BRME1 to form a V-shaped assembly. Here it would be nice to show a SDS-PAGE gel for Figure 6a (or a gel from a similar experiment). I ask for this because it would be helpful to show that the MEILB2-BRME1 complex doesn't undergo (an albeit unlikely) major structural rearrangement that could account for a different interpretation of the SEC-MALS data. Could the authors show the molecular mass of a complex formed from the addition of a larger BRCA2 construct (e.g. MBP-BRCA2). Additionally, how well does the 161 kDa measured value match the theoretical size of the 2:4:4 complex? Figure 6c needs a scale bar, and possibly some additional annotation to help orientate the reader.

4. We have included SDS-PAGE of the elution fractions that correspond to the BRCA2-MEILB2-BRME1 and MEILB2-BRME1 runs from Figure 6a as Supplementary Figure 8a,b.

We have also analysed the fusion BRCA2-MEILB2-BRME1 complex, in which both MEILB2 and BRCA2 have N-terminal MBP tags. Its experimental molecular weight is 451 kDa, and the theoretical mass of the 2:4:4 fusion complex is 454 kDa. The theoretical mass of the untagged 2:4:4 complex is 176 kDa, which is close to the experimental value of 161 kDa. The fusion SEC-MALS data nicely support the untagged data, and we have added these to the manuscript as Supplementary Figure 8c (line numbers 279-281). In all cases, the theoretical masses are included in figure legends.

As requested, we have added scale bars to Figure 6c. To aid the reader, we have added a 3D reconstruction of the complex from the cryo-EM data as Figure 6d, which confirms the angle and dimensions of the structure. We have also added further description in the text of the 2D class averages and the 3D ab initio model (line numbers 293-300).

BRCA2-MEILB2-BRME1 acts as a DNA clamp. As above, the authors should show some additional characterisation of DNA binding using other approaches. In addition a useful control here would be what happens if one uses a mutant BRCA2 that cannot bind to the MEILB2-BRME1 complex? For the complexes that don't enter the gel, the authors should also acknowledge that these could be protein-DNA aggregates, rather than an organised network. The authors also suggest that the two beta-caps of the ternary complex interact cooperatively with a long DNA molecule. If this is true, the authors should quantitate their EMSAs and compare the Hill Coefficients used for a BRCA2-MEILB2 K26E BRME1 ternary complex on DNA versus one with a mutation that prevents BRCA2 mediated complex dimerisation. Furthermore, if the ternary complex binds DNA more tightly than the MEILB2-BRME1 complex, then might this not be SEC stable?

5. As outlined in response number 3, analysis of DNA-binding was limited by the technical restriction that DNA induces the precipitation of the MEILB2-BRME1 complex. This was even more challenging for the BRCA2-MEILB2-BRME1 complex, which was stable in high salt (>300 mM KCl) but precipitated at a physiological salt concentration of 150 mM KCl even in the absence of DNA. This is shown in the following image in which the protein is visualised at 325 mM and 150 mM KCl.

Meilb2 22-332 + Brme1 545-583 + Brca2 2232-2289

1. 20mM HEPES pH 7.5 + 325mM KCl
2. 20mM HEPES pH 7.5 + 150mM KCl

This precipitation prevented us from analysing DNA-binding in solution by approaches such as FP, ITC and SEC, and led to too high background binding for biotinylated DNA pull-downs. Instead, EMSAs provided a suitable means for analysing DNA-binding by setting up reactions at 325 mM KCl and then relying on salt reduction as the complex enters the gel and transferred into the TBE running buffer. This appeared to stabilise complexes and allowed analysis of a range of species between discrete structures and assemblies. As outlined in response number 3, we additionally analysed DNA-binding by BRCA2-MEILB2-BRME1 by atomic force microscopy (Figure 7h and Supplementary Figure 13), in which we directly visualised the induction of tangles and loops in a manner consistent with our model in which the V-shaped complex tethers together DNA molecules. We have added a discussion of the issue of precipitation of BRCA2-MEILB2-BRME1 in the manuscript (line numbers 309-313)

We agree that the nature of the large structures that do not enter the gel is uncertain, and their interpretation is complicated by the issue of BRCA2-MEILB2-BRME1 precipitation at 150 mM KCl. We envisage that the large resolving species are likely to be protein-DNA networks in which BRCA2-MEILB2-BRME1 bridges between DNA molecules as these not form in DNA-binding mutants. However, the larger species that do not enter the gel may be more complicated structures, such as a combination of bridging interactions and the process responsible for precipitation in solution. We have substantially edited our presentation of DNA-binding and the species formed by the BRCA2-MEILB2-BRME1 complex (and our use of BRME1 Δ RKTK), in which we include a discussion of precipitation and the possible nature of the complexes that we are visualising (line numbers 309-344).

As requested, we have quantified the EMSAs for the BRCA2-MEILB2 K26E-BRME1 complex. We cannot use a BRCA2 mutant that does not bind to MEILB2-BRME1 as all of the complexes were prepared by co-expression and co-purification, so this would

simply be the same as analysing MEILB2-BRME1 in isolation. Instead, as the intention would be to differentiate between cooperative binding by the two arms of the complex, we quantified EMSAs for the wild-type BRCA2-MEILB2-BRME1 binding to 75 bp and 300 bp DNA, in which binding by both arms should only be possible for the longer substrate. The K_D values and Hill coefficients are given below:

BRCA2-MEILB2-BRME1 with 75 bp DNA

$K_D = 400 \pm 40$ nM

$n = 4 \pm 1.6$

BRCA2-MEILB2-BRME1 with 300 bp DNA

$K_D = 310 \pm 40$ nM

$n = 4 \pm 1.6$

BRCA2-MEILB2 K26E-BRME1 with 300 bp DNA

$K_D = 300 \pm 20$ nM

$n = 3 \pm 0.7$

Hence, we don't see any clear differences between these experiments. Whilst quantification of EMSAs provides a useful estimate of K_D , it is likely that the measurement does not have sufficient precision to detect small differences in Hill coefficient that would be necessary to infer differences in cooperativity. Further, we may be observing two different types of cooperativity – that between the two arms binding to the same DNA molecule, and that of multiple complexes binding to the same DNA molecule (as is suggested by precipitation in solution). It would be very difficult to distinguish between these experimentally, especially through quantification of EMSAs. Nevertheless, we have included these data in Supplementary Figure 11c.

Minor points

Page 11, first paragraph. Figure 6a rather than Figure 5a

6. Thank you, we have corrected this error (line numbers 278-279).

Figure 5c, odd looking box around gel label

7. This was an artefact of pdf rendering, and has been corrected in the current submission.

Figure 5d, please use either nt or bp for the DNA length.

8. We have changed ssDNA to bases and dsDNA to base pairs.

Page 12, second paragraph. Wouldn't 75 bp B-form DNA have an approximate length of 25.5 nm rather than 22 nm? Assuming 3.4 Å rise per bp.

9. Yes, 75bp should be approximately 25.5 nm. We have corrected this in the text (line numbers 336-338).

Supplementary Figure 4b - please somehow make the colouring of the structures clear in the figure (e.g. make the label PDB 7LDG red)

10. We have modified the labels to include the colours of the two PDB models.

Reviewer #2 (Remarks to the Author):

BRCA2 is a major player in the control of HR and a few years ago several studies identified two proteins MEILB2 (also known as HSF2BP) and BRME1 (aka MEIOK21, MMR, C19ORF57 and 2930432K21Rik) as directly involved in BRCA2 activity. They interact with BRCA2 and somehow promote its loading on DNA. Previous studies have identified interactions between MEILB2 and BRCA2 and between MEILB2 and BRME1 and determined the structure of the BRCA2-MEILB2 complex (using the interacting protein domains). Further characterization of these proteins and their interactions is essential to understand how the ssDNA tails become accessible, and then loaded by the recombinases RAD51 and DMC1., and the challenge is to integrate the different partners involved.

*Here the authors gain insight into this direction by determining the structure of the MEILB2-BRME1 complex, then by adding the BRCA2 binding domain, they show a tetramerization of the complex including domains from all three proteins (BRCA2/BRME1/MEILB2). One of the new findings is that the MEILB2-BRME1 complex binds DNA, and the authors propose a function in bridging DNA helices when the complex is tetrameric. All these data are important, and necessary to gain insight into these proteins. As these proteins are truncated in several assays, it is difficult to determine what the *in vivo* behaviour will be. The BRCA2/BRME1/MEILB2 complex contains only a small domain of both BRME1 and BRCA2. This is a problem because BRCA2 has other domains with affinity for the DNA, which may strongly influence the overall properties of this complex. The author's interpretation is highly speculative and should therefore take this into account, as alternative scenarios are possible.*

11. As the reviewer highlights, the *in vitro* data used truncated constructs corresponding to small domains of BRME1 and BRCA2. This is necessary given the insolubility and very low yields of full-length BRME1 and BRCA2 proteins. Further, it is appropriate as these are conserved peptide regions within unstructured sequence, so are unlikely to be part of wider structured, and is consistent with other biochemical studies in the field. Given the low nano-molar affinity of the MEILB2-BRME1 and MEILB2-BRCA2 interactions (Ghouil et al 2021 and 2023; <https://www.nature.com/articles/s41467-021-24871-6> and <https://www.science.org/doi/10.1126/sciadv.adi7352>), and the stability and monodisperse nature of the 2:2 and 2:4:4 complexes, it appears unlikely that other regions of BRME1 and BRCA2 would interfere with these oligomers. Nevertheless, it is not possible for us to exclude this possibility. We envisage that the V-shaped structure likely combines with BRCA2's proximal DMC1-binding and DNA-binding domains to provide the coordinated architecture of recombination. We have added appropriate caveats throughout the manuscript and have added a paragraph addressing this issue in the discussion (line numbers 392-400).

Of note, a similar study has been very recently published: Ghouil, R. et al. (2023). BRCA2-HSF2BP oligomeric ring disassembly by BRME1 promotes homologous recombination. Sci Adv 9, eadi7352. 10.1126/sciadv.adi7352. In this study, the interactions are tested in different

ways (for instance MLEIB2 with BRCA2 but without BRME1, human proteins) thus leading to different observations. The authors should discuss one of the common observations, which is the tetrameric form of the BRCA2/BRME1/MEILB2 complex. It would be interesting to know if they have also observed the oligomeric form in the absence of BRME1.

12. Yes, this recently published paper reports the same oligomeric species for the BRCA2-MEILB2-BRME1 complex, and reports the 4:8 structure formed by BRCA2-MEILB2 in absence of BRME1, and its assembly into interlocked ring structures. We have also analysed the BRCA2-MEILB2 complex in isolation. It was prone to precipitation, which limited the data quality, but SEC-MALS indicated a molecular weight range (300-400 kDa) that is consistent with the 4:8 complex (theoretical mass – 313 kDa). We have included the SEC-MALS data as Supplementary Figure 14, and have added a paragraph to the discussion that describes the Ghouil et al paper and discusses the above points (line number 449-459).

A general comment on the presentation of the data is that often one does not know which protein domain is being tested, and these protein domains are inappropriately named as if they were full-length proteins, which is misleading. Thus, one has to search for methods or various parts of the text to get the information. This should be revised through the text, figures and legends.

13. Yes, we agree that this may be confusing. We have changed the name of the main BRME1 and BRCA2 constructs to BRME1_{MBD}, and BRCA2_{MBD}, and have changed the name of the crystallographic construct of MEILB2 to MEILB2 α . For simplicity, we have retained the use of MEILB2 for the main MEILB2 construct as this is almost full-length (22-338), deleting only the unstructured N-terminus. We have defined these early in the text and in Figure 2a.

Other comments

1) MEILB2-BRME1: Please check whether the domain is 22-81 or 22-79. Also, a comment is needed to understand why the first 21 aa have been removed (probably solubility issues), and what would be their predicted structure. This could be important information given the DNA interactions proposed in this study involving residues at the N ter of MEILB2.

14. The boundaries of the MEILB2 core construct are 22-81. We have ensured that this is correctly stated throughout the manuscript.

The first 21 amino-acids of MEILB2 are poorly conserved (Supplementary Figure 1a) and are predicted to be unstructured. They were removed to produce stable constructs of the structure for crystallization and biophysical analysis (in which flexible tails severely interfere with data quality). We have added a comment in the manuscript to confirm this point (line numbers 204-206). Further, please see below an AlphaFold2

model of the MEILB2-BRME1 2:2 complex using the full-length MEILB2 sequence, in which the N-termini are clearly predicted to be unstructured.

2) Fig3b and Sup Fig3: which fraction of the samples are loaded on the gels?

15. Figure 3b shows the amylose elution fractions following recombinant co-expression of MBP-MEILB2 with His-BRME1 (WT and 3E mutant). Supplementary Figure 4 shows all fractions from the same experiment (pellet, supernatant, and amylose flow-through, wash and elution). We have clarified this in the figure legend (line numbers 971-972).

3) Is the BRME1 3E mutant protein properly folded in vitro ?

16. We find that the BRME1 peptide is insoluble in absence of MEILB2. This is shown by the following gel in which BRME1 was co-expressed with MEILB2 51-122 (which lacks the BRME1-binding site), and the BRME1 peptide remained in the pellet.

Hence, it appears that BRME1 co-folds with MEILB2, and thus does not fold into a soluble form in isolation. Consequently, it is not possible to determine whether there are any differences between the wild-type and 3E mutant of BRME1 in isolation of MEILB2. Nevertheless, as the mutation replaces hydrophobic amino-acids with glutamate residues, it is highly unlikely that the mutation would have any negative effects on BRME1 solubility.

4) *DNA binding assays: which protein domains are used in the assays. Since the in vivo substrate is a dsDNA with ssDNA tails, the authors should test whether there is a specific affinity for ssDNA-dsDNA junctions. What is the so-called “random sequence DNA” of 75bp and 300bp?*

17. The DNA-binding assays are performed using the standard constructs of the manuscript, which are MEILB2 (22-338), BRME1_{MBD} (540-578) and BRCA2_{MBD} (2232-2283). This should be clearer as we have now adopted the above terminology of BRME1_{MBD} and BRCA2_{MBD}. We have clearly labelled cases in which we have used the MEILB2 K26E mutant, the BRME1_{MBD} R540E R549E mutant, and the C-terminal BRME1 truncation BRME1_{MBD} ΔRKTK.

We have tested binding of both MEILB2-BRME1 and BRCA2-MEILB2-BRME1 to an ssDNA-dsDNA junction, and observed similar a similar pattern of binding and no higher affinity than observed for dsDNA or ssDNA substrates. Hence, it does not appear that the complex specifically recognizes junction DNA. We have included these data as Supplementary Figures 7a and 11b.

We have added the sequences of the 75 bp and 300 bp dsDNA, and the ssDNA-dsDNA junction substrates to the methods (line numbers 569-581).

Do the mutations introduced (on K26 and R540-549) have any effect on the dimerization? The authors conclude that the DNA binding is a consequence of the complex formation. To conclude this, data about the DNA binding affinity of MEILB2 and BRME1 separately should be shown.

18. We have confirmed by SEC-MALS that the K26E and R540E R549E mutants do not affect the ability of MEILB2-BRME1 to form stable 2:2 complexes. We have added these data as Supplementary Figure 7b,c (line numbers 259-261).

We have confirmed that isolated MEILB2 cannot bind DNA, and have included these data as Supplementary Figure 7d (line numbers 266-267). We have further confirmed that the BRCA2-MEILB2 complex is largely incapable of DNA-binding (with some residual binding at high concentration), and have included these data as Supplementary Figure 11d (line numbers 318-320). As outlined in response number 16, it is not

possible to test DNA-binding by BRME1 as this peptide is insoluble in absence of complex formation with MEILB2. Together, these findings support the argument that DNA-binding is a consequence of complex formation.

5) Complex with BRCA2. The proposition of a V-shaped complex is potentially interesting however not supported by the data: on the CryoEM (fig6c), angles are quite variable. So, the claim that the angle is approximately 90° is not convincing. Quantitative measurements of the angle and the distance between the ends of the arms should be performed. Is the tetramer formed in the presence of K26E?

19. The proposed V-shaped structure, with an approximately 90° angle between its limbs, is based on the previously reported crystal structure of the BRCA2-MEILB2 ARM domain 2:4 complex in which the angulation between stems of coiled-coils is clearly observed. In this manuscript, we extended this structure by adding in the full MEILB2-BRME1 complexes (modelled from our core crystal structure), to provide a proposed model of the full complex. We validate this model through direct visualisation of a V-shaped structure by cryo-EM (we address the reviewer's concerns regarding cryo-EM data below) and the close fitting of the V-shaped ternary complex model to SEC-SAXS data. Hence, the model is based on the geometries observed in crystal structures and is validated by two distinct experimental methods. Thus, we believe that the proposed V-shaped model is well supported by experimental data.

The reviewer raises concerns regarding the angles observed in the cryo-EM 2D class averages (Figure 6c). In cryo-EM, we observe 2D projections of 3D objects, so the orientation of the object substantially distorts the appearance of the 2D projection. In this case, the only way in which the angle between arms of the V-shaped complex could be measured directly from the 2D class is if the structure happens to be lying flat on the grid. In any other orientation, the angle will be distorted by the rotation of the object. Hence, the different angles observed in the 2D classes are likely due to different orientations of the same object rather than the same views of different objects. To illustrate this, we have included some of the classes below alongside orientations of the same V-shaped 2:4:4 complex model that may explain the projections.

Further, we have constructed a 3D *ab initio* model from the cryo-EM data, confirming that the images are consistent with a single V-shaped structure in which the arms are at approximately right-angles to one another. We have included the 3D model in Figure 6d.

The reviewer requests that we make quantitative measurements of the distance and angle between the arms. As outlined above, it is not possible to take measurements from individual cryo-EM images or 2D classes as they are projections of 3D objects at various orientations. Indeed, angles in a 2D projection may appear larger or smaller than that of the 3D object, and distances in a 2D projection may appear smaller than in the 3D object, depending on its orientation. Instead, we have taken measurements of the 3D *ab initio* model, indicating a distance of 25 nm between the ends of the arms and an angle of approximately 100° between the two arms. In the V-shaped 2:4:4 complex model, we measure the same distance of 25 nm between the ends of the arms, and a slightly smaller angle of 95° between the two arms. These measurements are very similar, indicating that the model is consistent with the experimental cryo-EM data. We have added the measurements to the cryo-EM 3D model (Figure 6d), we have updated

the measurement in the model to 95° (Supplementary Figure 9). Further, we have updated the description in the text (line numbers 293-299), and for accuracy, we have updated our conclusion in the text to state that the V-shaped structure has an angle of slightly greater than 90° between arms.

Yes, the BRCA2-MEILB2-BRME1 2:4:4 complex is retained in the MEILB2 K26E mutant. We have added the relevant SEC-MALS data in Supplementary Figure 12c (line numbers 352-353).

The model of bridging the two ends is not convincing. The first point is that there is no need to bridge the two ends for repair by HR, on the contrary, current models propose that the two ends act differently, one searching for the homolog, and the other potentially interacting with the sister chromatin. The only scenario where one does expect a bridging of two ends is in the NHEJ pathways, which is not the case here. Bridging homologs or sister chromatids may be more relevant.

20. We agree that bridging between homologues is far more likely than bridging between the two ends for precisely the reasons outlined by the reviewer. We chose to mention both possibilities as it is not possible to distinguish between these experimentally, and the mechanics of meiotic recombination are poorly understood so it is possible that a role for end tethering could become apparent in future. We have edited the text and the legend of Figure 8 to make it clear that bridging between homologues is the most likely role for DNA-tethering by the V-shaped structure (line numbers 406-407, 409 and 1067-1068).

Others

P3: typo: 3' single-stranded DNA

21. Thank you, we have corrected this error (line number 57).

P4: binds together homologous chromosome axes.

22. We have made this change (line number 71).

P11: Figure 5a: should be 6aBRME1 RGTK deletion: does it affect MEILB2 binding?

23. That is correct – we have made this change. The RGTK deletion does not affect MEILB2-binding. We have added in SEC-MALS data and an SDS-PAGE of the SEC elution of the MEILB2 core-BRME1_{MBD} ΔRGTK as Supplementary Figure 12a,b, which confirm that it is a stable 2:2 complex (line numbers 323-324).

Reviewer #3 (Remarks to the Author):

In this manuscript, Gurusaran et al. studied the structure of MEILB2-BRME1 complex with or without BRCA2 binding. They found that MEILB2-BRME1 complex binds both ssDNA and dsDNA with its N-terminal-cap. The ARM domain of MEILB2, which locates on its C-termini, binds to the MBD domain of BRCA2 and dimerize to form a V-shaped DNA clamp. With these findings, they proposed that the BRCA2-MEILB2-BRME1 complex may function as a DNA clamp to connect the resected DNA ends or homologous chromosomes to facilitate meiotic recombination. The results and structure analysis of MEILB2-BRME1 complex are reliable and of certain significance. However, several points need to be concerned:

Major concerns:

1. The main problem for this research is that all the conclusions and proposals are based on in vitro results. The proposed structure of BRCA2-MEILB2-BRME1 complex could not answer how it facilitates RAD51 and DMC1 loading. Also, there is no evidence to show whether the proposed function of BRCA2-MEILB2-BRME1 based on its V-shaped DNA clamp structure exists in meiosis or not. In fact, Brca2 Δ 12/ Δ 12 mice, whose BRCA2 does not interact with MEILB2, showed no defects in HR, indicating that the proposed function of BRCA2-MEILB2-BRME1 based on its V-shaped DNA clamp structure is dispensable in meiosis.

24. The reviewer correctly highlights that this is a largely *in vitro* study (although we do include mutagenesis that tests the MEILB2-BRME1 interface *in vivo*), so findings are mostly based on biochemical evidence. This is important as biochemical and structural studies *in vitro* are the only means by which we can obtain truly molecular information regarding the structures and functions of individual proteins and complexes.

The reviewer comments that we do not answer the question of how BRCA2-MEILB2-BRME1 facilitates RAD51/DMC1 loading. This is correct, and we do not claim to have answered this question. It remains unknown how BRCA2 facilitates RAD51 loading despite its prominence and following decades of study. Hence, whilst we still lack the most fundamental structural and functional information regarding RAD51 loading by BRCA2, it will not be possible to determine how MEILB2-BRME1 affects this process.

The reviewer comments that there is no evidence regarding whether the structure exists *in vivo*. This is correct, and is also the case for most biological molecules as it is not yet technically possible to resolve the structures of protein complexes within cells. This is particularly challenging in recombination as proteins have a low abundance, and is further complicated in meiosis by the difficulty of genetic manipulation and the relatively small quantities of tissue. Hence, it is not possible to perform basic analysis such as gel filtration and western blotting of cell lysate to assess whether a protein complex changes sizes between a wild-type and knockout/mutant samples.

The reviewer cites the *Brca2* Δ 12/ Δ 12 data as evidence that the V-shaped clamp structure is dispensable in meiosis. However, this statement is misleading. The

interaction between BRCA2 MBD and MEILB2 requires two BRCA2 acidic residues, D2294 and D2317, and single mutations, either D2294R or D2317R, were not sufficient to mislocalise BRCA2 from recombination nodules in murine spermatocytes (Fig. 4A of <https://www.nature.com/articles/s41594-021-00635-0>). The *Brca2* Δ 12/ Δ 12 mice only lack D2294 but retain D2317, suggesting that the BRCA2-MEILB2 interaction was not abolished. This was acknowledged by the authors reporting the *Brca2* Δ 12/ Δ 12 mice, showing the residual interaction of truncated BRCA2 and MEILB2/HSF2BP in the *Brca2* Δ 12/ Δ 12 mice testes, as demonstrated by MEILB2/HSF2BP IP (Fig. 6C of <https://www.nature.com/articles/s41467-021-24871-6>). Rather, complete mislocalization of the BRCA2 D2294R/D2317R double mutant (Fig. 4A of <https://www.nature.com/articles/s41594-021-00635-0>) has established that the MEILB2-BRCA2 interaction established in our previous and recent study is key for the *in vivo* localization of these proteins.

2. Though MEILB2-BRME1 could bind DNA directly, their recruitment requires MEIOB and SPATA22, indicating that these proteins may interact with each other when they are recruited. Thus, the presence of MEIOB and SPATA22 could also change the structure of BRCA2-MEILB2-BRME1 complex *in vivo*, which should be considered when analyzing the structure and function of BRCA2-MEILB2-BRME1 complex.

25. As the reviewer states, MEIOB and SPATA22 are required for MEILB2-BRME1 recruitment *in vivo*. However, no published data exist regarding whether the proteins interact. Hence, it remains unknown whether MEIOB/SPATA22 bind to MEILB2-BRME1 directly or whether they are simply essential components of the upstream pathway. It appears unlikely that a hypothetical MEIOB/SPATA22 interaction could interfere with either the MEILB2-BRME1 or MEILB2-BRCA2 interfaces as these have low nano-molar affinity (<https://www.nature.com/articles/s41467-021-24871-6> and <https://www.science.org/doi/10.1126/sciadv.adi7352>), and the consequent 2:2 and 2:4:4 complexes are highly stable and monodisperse in solution. It is more likely that any binding would be mediated by the wider sequence of BRME1, which appears to consist of a series of peptide binding sites within intrinsically disordered sequence. We have attempted to purify MEIOB and SPATA22 to test whether they interact with MEILB2-BRME1. However, both proteins have proven stubbornly insoluble, whether expressed separately or together, so it is not yet technically possible to test the proposed interaction. Hence, for the above reasons, it is unlikely that a MEIOB/SPATA22 interaction would affect the structure, but we cannot formally exclude the possibility that this or any other hypothetical interaction could affect the BRCA2-MEILB2-BRME1 structure. We have commented on the potential interaction with MEIOB/SPATA22 and the possible role of the 2:4:4 complex within a wider ternary complex in the discussion (line numbers 417-429 and 461-471).

3. *BRCA2-MEILB2-BRME1 formed a 2:4:4 complex when BRCA2-MBD is added. However, BRCA2 is a large protein that contained 3148aa rather than the 52aa BRCA2-MBD used in this research. Could it be possible that BRCA2-MEILB2-BRME1 complex forms 1:2:2 rather than 2:4:4 structure at the presence of full length BRCA2?*

26. As the reviewer highlights, the structural and biochemical experiments were performed using small MBD of BRCA2 rather than the full-length protein. It appears unlikely that the 2:4:4 BRCA2-MEILB2-BRME1 structure would be affected by the presence of the wider BRCA2 sequence for the following reasons. Firstly, the MBD is peptide-binding site located in unstructured sequence (rather like the BRC repeats, exon 14 and exon 27 that bind to RAD51 and DMC1), so is unlikely to be part of a wider structure. Secondly, the interaction between MEILB2 and BRCA2's MBD has low nano-molar affinity so would be very difficult to disrupt (<https://www.nature.com/articles/s41467-021-24871-6>). Thirdly, the nature of the binding interface requires the presence of two opposing MEILB2 dimers and it is not sterically possible for both interfaces of BRCA2's MBD to bind to the same MEILB2 dimer, so 2:4:4 complex formation is essentially encoded in the binding mode (<https://www.nature.com/articles/s41594-021-00635-0>). Finally, the 2:4:4 BRCA2-MEILB2-BRME1 structure is extremely stable and mono-disperse, and we do not detect the presence of any 1:2:2 complex in solution. Ideally, we would of course confirm the 2:4:4 structure using full-length BRCA2. However, this is not technically possible as existing expression and purification systems for BRCA2 have a yield of ~500 ng from 20 l cultures (<https://www.nature.com/articles/nsmb.1905>), whereas at least 200 µg is required for each SEC-MALS experiment. Indeed, it for the same technical reasons that the oligomeric state of BRCA2 in isolation and upon RAD51/DMC1-binding remains uncertain, and that all published research on MEILB2-binding has been performed using BRCA2's MBD rather than the full-length protein. We have added a paragraph in the discussion that addresses this issue (line numbers 392-400), and discuss the potential relationship between the 2:4:4 and BRCA2 structure in a later section of the discussion (line numbers 431-447).

4. *It seemed that MEILB2-BRME1 binds both ssDNA and dsDNA. Given that the known function of BRCA2-MEILB2-BRME1 is to facilitate the replacement of RPA by RAD51 and DMC1, which takes place on ssDNA, the authors should pay more attention on the interaction of BRCA2-MEILB2-BRME1 with ssDNA or RPA coated ssDNA, rather than dsDNA. Taken together, the significance of this research is limited.*

27. We have tested DNA-binding by MEILB2-BRME1 and BRCA2-MEILB2-BRME1 using ssDNA, dsDNA and ssDNA-dsDNA junctional substrates. In all cases, we observed similar patterns of binding and affinities, suggesting that binding is mediated through the backbone rather than being specific for a certain substrate. These data are provided in Figures 5d-f and 7a, and Supplementary Figures 7a and 11a,b. Given this observation, we chose to use dsDNA as the predominant substrate throughout the manuscript as it has several technical advantages. Most notably, it is possible to study

dsDNA substrates of any length without concerns regarding formation of secondary structure. In contrast, to avoid ssDNA forming secondary structure requires the use of polydT substrates (which tend to be limited to lengths of around 100 bases) or the use of random ssDNA of any length in the presence of RPA. Whilst the latter option would be an interesting experiment in its own right, it would be testing two different questions at once – whether/how the BRCA2-MEILB2-BRME1 complex binds DNA and whether it can displace RPA. Further, this is complicated by the requirement for MEIOB-SPATA22 in meiosis, which may be necessary for a displacement role of MEILB2-BRME1, and BRCA2's DBD may be required for displacement of RPA. As outlined in response numbers 25 and 26, significant technical challenges need to be overcome before we can study all of these full-length proteins in the same biochemical/biophysical system. Hence, we chose to use dsDNA as a way of studying DNA-binding in isolation. We comment on the choice of DNA substrate in the results section (line numbers 246-253) and we address the potential wider role of RPA and MEIOB-SPATA22 in the discussion (line numbers 417-429).

REVIEWERS' COMMENTS

Reviewer #1 (Remarks to the Author):

The authors have addressed all of my comments and concerns. As such I am fully supportive of publication of the manuscript in its current form.

Reviewer #2 (Remarks to the Author):

The authors have answered all my comments and questions. The paper is suitable for publication.

Please note one typo: L462: change 5'-ssDNA to 3'-ssDNA

Reviewer #3 (Remarks to the Author):

I have read the revised manuscript and rebuttal letter, and satisfy with the authors' answers to my questions 3 and 4. However, some concerns are still existed for question 1.

For question 1, "The main problem for this research is that all the conclusions and proposals are based on in vitro results. The proposed structure of BRCA2-MEILB2-BRME1 complex could not answer how it facilitates RAD51 and DMC1 loading. Also, there is no evidence to show whether the proposed function of BRCA2-MEILB2-BRME1 based on its V-shaped DNA clamp structure exists in meiosis or not. In fact, *Brca2* Δ 12/ Δ 12 mice, whose BRCA2 does not interact with MEILB2, showed no defects in HR, indicating that the proposed function of BRCA2-MEILB2-BRME1 based on its V-shaped DNA clamp structure is dispensable in meiosis." When asking this question, our major concern is that the conclusions of this manuscript is based on in vitro experiments or studies in cultured cells which do not require MEILB2 or BRME1 for DSB repair. While the proposed function of the proteins in this manuscript has no in vivo evidence either. For these two reasons, I think the reliability of the conclusions drawn in this manuscript should be considered cautiously.

We do understand, as the authors pointed out, studying the structures of protein complexes in vivo is difficult, while understanding how MEILB2 and BRME1 functions together with BRCA2 to execute their known function (RAD51 and DMC1 recruitment) is hard either. However, the authors need to provide evidence for the suggested function of BRCA2-MEILB2-BRME1 complex does exist and is essential for (or at least involved in) HR repair during meiosis. For example, by generating MEILB2 or BRME1 mutant mice that interrupt the DNA clamp formation but do not disturb RAD51/DMC1 recruitment, and show if they have defects in DSB repair. Or, at least, if the suggested function of BRCA2-MEILB2-BRME1 existed in meiosis, they should provide solid evidence for MEILB2-BRME1 localization on RAD51/DMC1 coated ssDNA, which contained no RPA (as suggested by Mol Cell. 2020 Aug 20;79(4):689-701, RPA is almost absent on RAD51/DMC1 coated ssDNA during homology search) (for example, by showing that a certain ratio of MEILB2-BRME1 co-localized with RAD51/DMC1 but not RPA under super-resolution microscope).

Also, for the *Brca2* Δ 12/ Δ 12 mice, we do agree that there is residual interaction of truncated BRCA2 and MEILB2/HSF2BP in the *Brca2* Δ 12/ Δ 12 mice testes. However, as the major interaction between BRCA2 and MEILB2/HSF2BP is disturbed by the truncation, it is also be

possible that the DNA clamp structure is disturbed due to the loss of D2294. This should be tested by inducing D2294 mutation in the MBD in this manuscript and discussed.

Response to reviewers' comments
'MEILB2-BRME1 forms a V-shaped DNA clamp upon BRCA2-binding in
meiotic recombination'
(NCOMMS-23-48925A)

Reviewer #1 (Remarks to the Author):

The authors have addressed all of my comments and concerns. As such I am fully supportive of publication of the manuscript in its current form.

1. No changes required.

Reviewer #2 (Remarks to the Author):

The authors have answered all my comments and questions. The paper is suitable for publication.

Please note one typo: L462: change 5'-ssDNA to 3'-ssDNA

2. We have made this correction.

Reviewer #3 (Remarks to the Author):

I have read the revised manuscript and rebuttal letter, and satisfy with the authors' answers to my questions 3 and 4. However, some concerns are still existed for question 1.

For question 1, "The main problem for this research is that all the conclusions and proposals are based on in vitro results. The proposed structure of BRCA2-MEILB2-BRME1 complex could not answer how it facilitates RAD51 and DMC1 loading. Also, there is no evidence to show whether the proposed function of BRCA2-MEILB2-BRME1 based on its V-shaped DNA clamp structure exists in meiosis or not. In fact, Brca2 Δ 12/ Δ 12 mice, whose BRCA2 does not interact with MEILB2, showed no defects in HR, indicating that the proposed function of BRCA2-MEILB2-BRME1 based on its V-shaped DNA clamp structure is dispensable in meiosis." When asking this question, our major concern is that the conclusions of this manuscript is based on in vitro experiments or studies in cultured cells which do not require MEILB2 or BRME1 for DSB repair. While the proposed function of the proteins in this manuscript has no in vivo evidence either. For these two reasons, I think the reliability of the conclusions drawn in this manuscript should be considered cautiously.

We do understand, as the authors pointed out, studying the structures of protein complexes in vivo is difficult, while understanding how MEILB2 and BRME1 functions together with BRCA2 to execute their known function (RAD51 and DMC1 recruitment) is hard either. However, the authors need to provide evidence for the suggested function of BRCA2-MEILB2-BRME1 complex does exist and is essential for (or at least involved in) HR repair during meiosis. For

example, by generating MEILB2 or BRME1 mutant mice that interrupt the DNA clamp formation but do not disturb RAD51/DMC1 recruitment, and show if they have defects in DSB repair. Or, at least, if the suggested function of BRCA2-MEILB2-BRME1 existed in meiosis, they should provide solid evidence for MEILB2-BRME1 localization on RAD51/DMC1 coated ssDNA, which contained no RPA (as suggested by Mol Cell. 2020 Aug 20;79(4):689-701, RPA is almost absent on RAD51/DMC1 coated ssDNA during homology search) (for example, by showing that a certain ratio of MEILB2-BRME1 co-localized with RAD51/DMC1 but not RPA under super-resolution microscope).

Also, for the Brca2 Δ 12/ Δ 12 mice, we do agree that there is residual interaction of truncated BRCA2 and MEILB2/HSF2BP in the Brca2 Δ 12/ Δ 12 mice testes. However, as the major interaction between BRCA2 and MEILB2/HSF2BP is disturbed by the truncation, it is also possible that the DNA clamp structure is disturbed due to the loss of D2294. This should be tested by inducing D2294 mutation in the MBD in this manuscript and discussed.

3. In accordance with the editorial decision to overrule the request for further mouse in vivo work, we have added caveats and limitations to the discussion.